# Structural insights into terminal arabinosylation of mycobacterial cell wall arabinan

Yaqi Liu [1], Chelsea M. Brown [2,3], Satchal Erramilli [4], Yi-Chia Su[5], Shih-Yun Guu[5], Po-Sen Tseng [6], Yu-Jen Wang[5], Nam Ha Duong [5,7,8], Piotr Tokarz[4], Brian Kloss [1], Cheng-Ruei Han[5], Hung-Yu Chen[5], José Rodrigues [9], Kay-Hooi Khoo[5,10], Margarida Archer [9], Anthony A. Kossiakoff[4], Todd L. Lowary [5,6,10] ✉, Phillip J. Stansfeld [2] ✉, Rie Nygaard [1,11] ✉ & Filippo Mancia [1] ✉

The global challenge of tuberculosis, caused by *Mycobacterium tuberculosis (Mtb)*, is compounded by the emergence of drug-resistant strains. A critical factor in *Mtb*'s pathogenicity is its intricate cell envelope, which acts as a formidable barrier against immune defences and pharmacological interventions. Central to this envelope are arabinogalactan (AG) and lipoarabinomannan (LAM), two complex polysaccharides containing arabinan domains essential for maintaining cell wall structure and function. The arabinofuranosyltransferase AftB plays a pivotal role in the biosynthesis of these arabinan domains by catalyzing the addition of β-(1 → 2)-linked terminal arabinofuranose residues. Here, we present the cryo-EM structures of *Mycobacterium chubuense* AftB in both its apo form and bound to a donor substrate analog, resolved at 2.9 Å and 3.4 Å resolution, respectively. These structures reveal that AftB has a GT-C fold, with a transmembrane (TM) domain comprised of eleven TM helices and a periplasmic cap domain. AftB has a distinctive irregular, tube-shaped cavity that connects two proposed substrate binding sites. Through an integrated approach combining structural analysis, biochemical assays, and molecular dynamics simulations, we delineate the molecular basis of AftB's reaction mechanism and propose a model for its catalytic function.

Tuberculosis (TB), caused by *Mycobacterium tuberculosis (Mtb)*, is the world's second deadliest infectious disease, surpassed only by COVID-19 during the recent pandemic[1]. The resilience and virulence of *Mtb* are largely due to its unique cell envelope—a lipid-rich fortress that confers resistance to host immune mechanisms and provides an impenetrable barrier to a wide range of antibiotics[2–5]. This complex cell envelope is comprised of three major components: long-chain mycolic acids that form a protective lipid layer, peptidoglycan that provides cell shape and structural rigidity, and arabinogalactan (AG), which links these two elements (Fig. 1a). AG is a polysaccharide with a linear galactan backbone and a highly branched arabinan domain. Together, these three components constitute the mycolyl–arabinogalactan–peptidoglycan (mAGP) complex, an impermeable shield critical to *Mtb*'s survival and pathogenicity[6–9] (Fig. 1a). In addition to the mAGP complex, the cell

A full list of affiliations appears at the end of the paper. ✉e-mail: tlowary@as.edu.tw; Phillip.Stansfeld@warwick.ac.uk; rin7007@med.cornell.edu; fm123@cumc.columbia.edu

**Fig. 1 | Biosynthetic pathway and structural architecture of AftB. a** Schematic of the arabinogalactan biosynthetic pathway in mycobacteria, featuring the enzymatic reactions carried out by AraTs. Enzyme names are indicated above the reaction arrows. The inset shows the β-(1 → 2) arabinosyl transfer reaction catalyzed by AftB. The arabinan chain is shown linked to the 10th position of the galactan chain through a β-(1 → 2) bond. While this linkage also occurs at the 8th or 12th position, the 10th position is depicted here for clarity. **b** Cryo-EM density map of AftB in complex with Fab-B3. AftB is shown in blue, while Fab-B3 is shown in maroon. **c** Topological diagram of AftB showing the arrangement of secondary elements in the TM region and in the PD, annotated and colored in correspondence with (**d**). The catalytic residue D62 is denoted as a red dot. **d** Structure of AftB shown in ribbon. The unmodeled region of AftB is marked as dashed lines, and membrane boundaries are marked by dotted horizontal bars.

envelope contains an array of glycolipids, including phosphatidylinositol mannosides (PIMs) and their structurally related derivatives: lipomannan (LM), lipoarabinomannan (LAM), and its delipidated form, arabinomannan (AM). These glycolipids play key roles in *Mtb*'s growth, virulence, and interaction with the host immune system[9,10]. Among them, LAM/AM stands out due to its intricate structure, comprising a mannan core with a highly branched arabinan domain[10–14] (Supplementary Fig. 1a).

The unique characteristics of the mycobacterial cell envelope have made it a pivotal target for anti-tuberculosis (TB) therapeutics[5,6,15–17]. Front-line drugs, such as isoniazid and ethambutol, specifically target *Mtb* cell envelope biosynthesis[18,19]. Isoniazid disrupts the synthesis of mycolic acids, a key component of the mAGP[18–20] complex, while ethambutol inhibits enzymes essential for arabinan biosynthesis, thereby impairing AG and LAM maturation[21–24]. Despite their efficacy, the rise of drug-resistant and multidrug-resistant *Mtb* strains has exposed the limitations of existing TB treatments and underscore the urgent need for innovative therapeutic targets and strategies[5,25,26].

The biosynthesis of the arabinan domains of AG and lipoarabinomannan LAM is catalyzed by a series of arabinofuranosyltransferases (AraTs)[5,6]. These membrane-embedded glycosyltransferases are responsible for transferring arabinofuranose (Ara*f*) units from β-D-arabinofuranosyl-1-monophosphoryldecaprenol (DPA), the only known D-arabinose donor substrate, onto growing polysaccharide chains[27–29]. For AG, the process begins with the α-(1 → 5)-arabinosyltransferase AftA, which attaches the initial α-D-Ara*f* residue to the mature galactan backbone[30] (Fig. 1a). The chain is then extended via α-(1 → 3)[24] linkages by the enzymes EmbA and EmbB. (Fig. 1a). For LAM, the priming enzyme responsible for initiating arabinan synthesis remains unidentified. Nonetheless, EmbC is involved in the extension of arabinan-primed LM with α-(1 → 5)-Ara*f* residues[31]. AftC introduces α-(1 → 3)-branching within both AG and LAM, adding structural complexity to these components of the cell envelope[32] (Fig. 1a). Recent studies suggest that AftD, previously thought to function like AftC[33], is instead an α-(1 → 5) AraT that extends α-(1 → 3) branched motifs in the arabinan domain[34]. Finally, AftB, an integral membrane enzyme encoded by the gene Rv3805c in *Mtb*, adds terminal β-(1 → 2)-linked D-Ara*f* residues to both AG and LAM/AM[35,36] (Fig. 1a and Supplementary Fig. 1a). In AG, this results in a terminal hexa-arabinofuranoside motif (Ara*f*$_6$) at the non-reducing end, which provides the anchoring point for mycolic acid[6,37] (Fig. 1a). In LAM/AM, AftB adds β-(1 → 2)-linked terminal-Ara*f* residues to both branched and linear arabinan motifs, with the latter resulting in linear Ara*f*$_4$ structures with terminal β-linked Ara*f* motifs[35] (Supplementary Fig. 1a). The AG is further ligated to peptidoglycan by the phosphotransferase Lcp1, completing the assembly[38] (Fig. 1a).

AftB plays a critical role in maintaining the integrity of the mycobacterial cell envelope and is a promising drug target[35,36,39] due to its essentiality[35,36] in *Mtb*. Despite its importance, the structural and molecular basis of AftB's function remains poorly understood.

Here, we present single-particle cryogenic-electron microscopy (cryo-EM) structures of *Mycobacterium chubuense* AftB in both apo and substrate-analog-bound states. By integrating structural data, biochemical assays, and molecular dynamics (MD) simulations, we identify key substrate-binding sites and propose a mechanism for AftB's enzymatic activity.

## Results

### Structure determination of AftB

We screened AftB orthologs from 45 mycobacterial species for expression in *E. coli* and identified *Mycobacteria chubuense* AftB (*Mc*AftB) as the most promising target for structural studies. To verify the activity of recombinant AftB, we adapted a previously reported arabinosyltransferase assay[40]. This assay uses a synthetic tetrasaccharide acceptor substrate (Supplementary Fig. 2a) corresponding to the non-reducing terminus of AftB's natural substrate in AG and LAM/AM, and farnesyl phosphoarabinose (FPA) as a donor

surrogate (Supplementary Fig. 2b). FPA, known to be an effective donor for mycobacterial AraTs, was chosen for its ease of preparation and handling compared to the native substrate, DPA[40,41]. Mass spectrometry analysis of the acetylated reaction products confirmed that AftB catalyzes the stepwise transformation of the tetrasaccharide into a pentasaccharide and, ultimately, a hexasaccharide, thus demonstrating the enzymatic activity of recombinant *Mc*AftB (Supplementary Fig. 2a–c and Supplementary Fig. 2e).

A previous study demonstrated AftB activity using a disaccharide acceptor[36]. To expand upon this, we opted for a larger, branched tetrasaccharide acceptor to investigate whether AftB exhibits a preference in the sequential addition of the two β-(1 → 2)-linked Ara*f* residues during the formation of Ara*f*$_6$. The resulting pentasaccharide and hexasaccharide products were analyzed using mass spectrometry (MS/MS) (Supplementary Fig. 3).

The permethylated Ara*f*$_5$ and Ara*f*$_6$ oligomers with an azide group were detected by MALDI-MS as $[M+Na]^+$, accompanied by prominent in source loss of an $N_2$ group (−28 Da). When the sodiated molecular ions were selected for Collision-Induced Dissociation (CID) MS/MS, all reducing end fragment ions detected, including the base peaks, similarly showed a loss of $N_2$. Of note, the fragmentation pattern afforded is identical to that observed previously when permethylated Ara oligomers were analyzed by MALDI TOF/TOF MS/MS[42,43]. This allows unambiguous assignment of the expected branching pattern of Ara$_5$ and Ara$_6$, as illustrated in Supplementary Fig. 3.

For the Ara*f*$_5$ product (Supplementary Fig. 3a), the loss of the third Ara*f* from the non-reducing end could not occur without creating another free OH ($m/z$ 472 instead of 486), whereas the loss of one ($m/z$ 806), two ($m/z$ 646) and four ($m/z$ 326) Ara*f* residues could result from a single glycosidic cleavage. This is consistent with a branched Ara*f*$_5$ structure extended by one and two Ara*f* residues from the branch point. The same applies to the Ara*f*$_6$ product (Supplementary Fig. 3b), whereby a single glycosidic cleavage could only lead to the loss of one ($m/z$ 966) and two ($m/z$ 806) but not three Ara*f* residues, which would require a second cleavage ($m/z$ 632 instead of 646). The branch point is further corroborated in both cases by the prominent sodiated B ion of Ara*f*$_2$ ($m/z$ 375) but not Ara*f*$_3$.

The sodiated B ion for Ara*f*$_4$ ($m/z$ 695) could additionally be detected for the Ara*f*$_5$ product, accompanied by the cross-ring cleavage ions, $^{2,4}A$ and $^{0,3}A$ ions at $m/z$ 765 and 737, respectively. The corresponding $^{2,4}A$ ion for Ara*f*$_3$ detected at $m/z$ 605 defines a O3,O5-di-Ara*f*-substitution. Importantly, the unique $^{0,3}A$ ions at $m/z$ 257 and 417 indicate that both isomeric Ara*f*$_5$ products were made. This is further supported by the ions at $m/z$ 297 and 457 resulting from further loss of the O3-substituent from the E ion (see the illustrated cleavage drawing in Supplementary Fig. 3a). Based on the similar intensity for the pairs of $m/z$ 257/297 versus $m/z$ 417/457, it can be further inferred that both isomeric products were approximately of equal abundance.

Finally, the $^{0,2}X$ ion, which would retain the O2-substituent, was detected at $m/z$ 862 for the Ara*f*$_5$ product. Because it could also be derived from the non-β-(1→2)-extended terminal Ara of either of the two isomeric Ara$_5$ products, it is by itself not critical evidence for terminal β-(1→2)-Ara*f*-substitution. However, in the case of the Ara*f*$_6$ product, the corresponding ion at $m/z$ 1022 unambiguously defines the O2-substitution (Supplementary Fig. 3b).

Insufficient quantities of the products were available for $^1H$ nuclear magnetic resonance (NMR) spectroscopy to confirm the β-stereochemistry of the added Ara*f* residues. However, the absence of reports describing α-Ara*f*-(1→2)-α-Ara*f* motifs in mycobacterial arabinan supports the conclusion that the products contained β-(1 → 2)-linked Ara*f* residues. Taken together, these data confirm that *Mc*AftB converts the tetrasaccharide acceptor into the native Ara*f*$_6$ motif present at the non-reducing terminus of AG and LAM/AM, and that the enzyme does not have a preference for which arm of the acceptor is first arabinofuranosylated.

The 78 kDa *Mc*AftB, was purified in detergent and reconstituted into lipid-filled nanodiscs for cryo-EM structure determination (Supplementary Fig. 1b, c). To provide fiducials for particle alignment and increase particle size to facilitate structure determination, we screened a synthetic phage display library to select recombinant antigen-binding fragments (Fabs) against *Mc*AftB[44,45]. Seven high-affinity Fabs were identified, with Fab-B3 selected for its strong binding affinity (Supplementary Fig. 1d, e).

Using 7164 micrographs of nanodisc-reconstituted apo *Mc*AftB in complex with Fab-B3, iterative 2D classification yielded high-quality class averages with distinct features for the transmembrane (TM) domain and bound Fab. Three-class ab initio particle sorting produced a 2.9 Å resolution map, enabling nearly complete atomic modeling of *Mc*AftB. (Supplementary Fig. 4a–c). This allowed us to build an almost complete atomic model of *Mc*AftB except for disordered regions comprising 28 residues at the N-terminus, 11 at the C-terminus, and a segment (323–334) in a cytoplasmic loop connecting two TM helices (8 and 9) (Fig. 1c, d and Supplementary Fig. 5). The Fab variable region was well resolved, allowing precise modeling of the *Mc*AftB–Fab interaction interface, while the disordered constant domain was excluded from the model (Fig. 1b and Supplementary Fig. 1f, g).

Additionally, we observed a distinct diacyl-glycerophospholipid-like density between TM helices 8 and 9. We tentatively attributed this to phosphatidylethanolamine (PE) (Supplementary Fig. 5), given its abundance in *E. coli* membranes[46].

## Overall structure of AftB

AftB comprises two domains: a TM domain with 11 α-helices and a C-terminal periplasmic domain (PD) composed of a combination of α-helices and β-strands (Fig. 1c, d). The TM helices of AftB are connected by five cytoplasmic loops (CL1–CL5) and four periplasmic loops (PL1–PL4). While most cytoplasmic loops are well resolved in the structure, CL4 (residues 323–334), linking TM helices 8 and 9, appears disordered in the structure (Fig. 1c, d).

Among the periplasmic loops, three loops (PL1, PL2, and PL3) have a more complex structure. PL1, between TM helices 1 and 2, includes two juxtamembrane (JM) helices, JM1 and JM2. PL2, connecting TM helices 3 and 4, contains JM3, while PL3, spanning between TM helices 7 and 8, contains JM4 and JM5 (Fig. 1c, d). The C-terminal PD of AftB extends from the end of TM helix 11, forming a dome-like structure that caps the TM helical bundle in the periplasmic space (Fig. 1b, d). This domain consists of two lobes: the first consists of four β-strands (β1–β4) and four α-helices (α1–α4), along with inter-linking loops. The second lobe contains seven inter-connected α-helices (α5–α11). PL5 serves as a linker, connecting TM11 to α1, and anchoring the PD to the TM bundle (Fig. 1d).

Fab-B3 binds specifically to the PD of AftB, with both its light and heavy chains contributing to the interaction interface (Supplementary Fig. 1f). The binding interface is characterized by a network of hydrogen bonds, predominantly mediated by the heavy chain of the Fab fragment (Supplementary Fig. 1g).

## Conserved GTC fold of AftB

Glycosyltransferases, a structurally diverse enzyme family, can be classified into three primary fold types: GT-A, GT-B, and GT-C. GT-A and GT-B folds typically feature Rossmann-like domains, while GT-C fold enzymes are characterized by multiple TM helices. GT-C enzymes are further divided into GT-$C_A$ and GT-$C_B$ subclasses[47–49]. Using the DALI server[50], we identified AftB as a member of the GT-$C_A$ superfamily of glycosyltransferases[47]. The first seven transmembrane helices of AftB adopt the GT-C glycosyltransferase fold[47,51]. Enzymes with the highest structural homology to AftB include bacterial oligosaccharyl-transferase PglB[52,53], bacterial aminoarabinose-transferase ArnT[54], and yeast glucosyltransferase ALG6[55] (Supplementary Fig. 7). These enzymes all belong to the GT-$C_A$ class[47] and share a conserved N-terminal module with AftB. This includes TM helices 1–7 and the two JM helices (JM1 and JM2) between TM helices 1 and 2.

While adopting the core GT-$C_A$ fold, AftB exhibits some notable differences. Specifically, both TM helices 3 and 10 in AftB are split into a short α-helix and a β-strand, forming a small anti-parallel β-sheet within the TM domain (Fig. 1c, d and Supplementary Fig. 5b). This feature is distinct to AftB and contrasts with other GT-$C_A$ enzymes, where TM3 typically exhibits a kink that splits it into two helices. The functional significance of these structural differences in AftB remains to be elucidated but could represent adaptations to its specific role in mycobacterial arabinan biosynthesis.

## The putative substrate cavity of AftB

A tube-shaped cavity was observed in AftB, spanning between the PD and TM domains. One end of the cavity opens towards the lipid membrane, and the other extends into the periplasmic space (Fig. 2a, b). This cavity appears to serve as the pathway for substrate entry. All GT-CA enzymes contain a conserved catalytically essential aspartate residue at the tip of JM helix 1[53–56]. In *Mc*AftB, this conserved aspartate corresponds to D62, which is part of a DD motif and is buried inside the cavity (Fig. 2a and Supplementary Fig. 8). A D62A mutant exhibited complete loss of enzymatic activity while maintaining comparable expression levels and proper folding compared to wild-type *Mc*AftB (Supplementary Fig. 9 and Supplementary Fig. 10), confirming its essential catalytic role.

The cavity is lined by JM helices 1, 2, and 4, as well as PL4, and connecting loops between TM and JM helices (Fig. 2a). On the periplasmic side, the cavity is defined by loops connecting TM helix 11 and α1, β3 and α4, β4 and α5, and α5 and α6. This architecture creates an entry point for the donor substrate, DPA, through the membrane-facing opening. This would permit the polar head of DPA to be oriented towards the catalytic core of the enzyme, with the lipid tail extending out into the lipid bilayer. The wider opening of the cavity towards the periplasm (Fig. 2a) presents a predominantly hydrophilic environment (Supplementary Fig. 11f), which could accommodate arabinan acceptor substrates with different end structures, enabling AftB to generate both the branched Ara$f_6$ terminal motif in AG and LAM/AM and the linear Ara$f_4$ terminal structures in LAM/AM.

To further substantiate our hypothesis for entry of the acceptor and donor substrates, we performed coarse-grained molecular dynamics (CG-MD) simulations to identify preferential binding sites and entry pathways for the substrates. We used the native donor substrate DPA (Supplementary Fig. 11a, left) and a hexasaccharide acceptor fragment, referred to as "terminal-Ara$f_4$" (Supplementary Fig. 11a, right). This acceptor species includes the branched motif present in the acceptor substrate used in the functional assay with two additional α-(1→5)-Ara$f$ residues at the non-reducing end (Fig. 1a, Supplementary Fig. 11a). DPA, initially randomly distributed in the membrane's upper leaflet, preferentially bound within the cavity defined by TM helices 5, 6, and 7, and JM helices 1, 2, and 4 (Fig. 2c), consistent with the cavity proposed above (Fig. 2a). Terminal-Ara$f_4$, starting in bulk solvent, rapidly associated with the cavity from the periplasmic side, binding near loops connecting β3 and α4, β4 and α5, and α5 and α6 (Fig. 2d). This corresponds to the predicted entryway for the acceptor substrate (Fig. 2a, b).

The presence of Fab-B3, used during structural determination, did not significantly alter substrate interactions, as shown by the CG-MD simulations (Supplementary Fig. 13a, right). Substrate binding was observed both with and without Fab. Furthermore, RMSF analyses of atomistic simulations revealed similar protein dynamics in apo, Fab-bound, and substrate-bound states (Supplementary Fig. 13a, left). These findings support that the Fab-bound structure represents a catalytically relevant conformation.

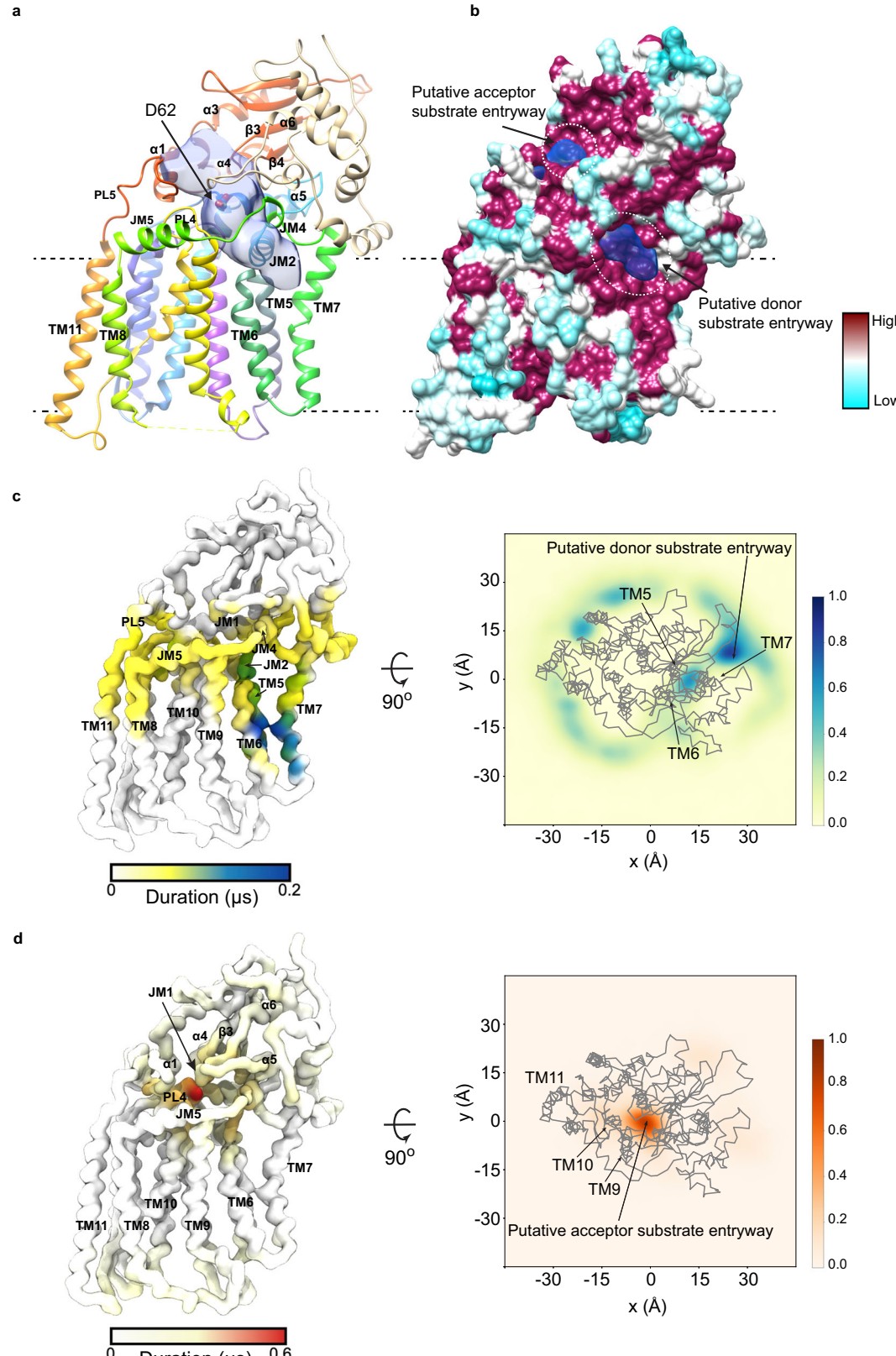

**Fig. 2 | Putative substrate-binding cavity. a** AftB structure shown as ribbon, and colored as in Fig. 1c, with the cavity shown in semitransparent blue. Volumes were calculated using the Voss Volume Voxelator server[114] with probe sizes of 2 Å and 8 Å. **b** Surface of AftB by conservation, with a gradient from magenta (absolutely conserved) to cyan (least conserved), cavity shown in semitransparent blue. Predicted entryways for substrates are marked with dotted circles. **c** Left shows the AftB backbone as a surface, colored by the duration of interaction with DPA in CG-MD simulations. The darker the color, the longer the duration of interaction. The right plot shows the density in the x and y dimensions of DPA relative to the protein shown in gray, where the darker regions show higher density. **d** Shows the same as (**c**) but for terminal-Ara$f_4$.

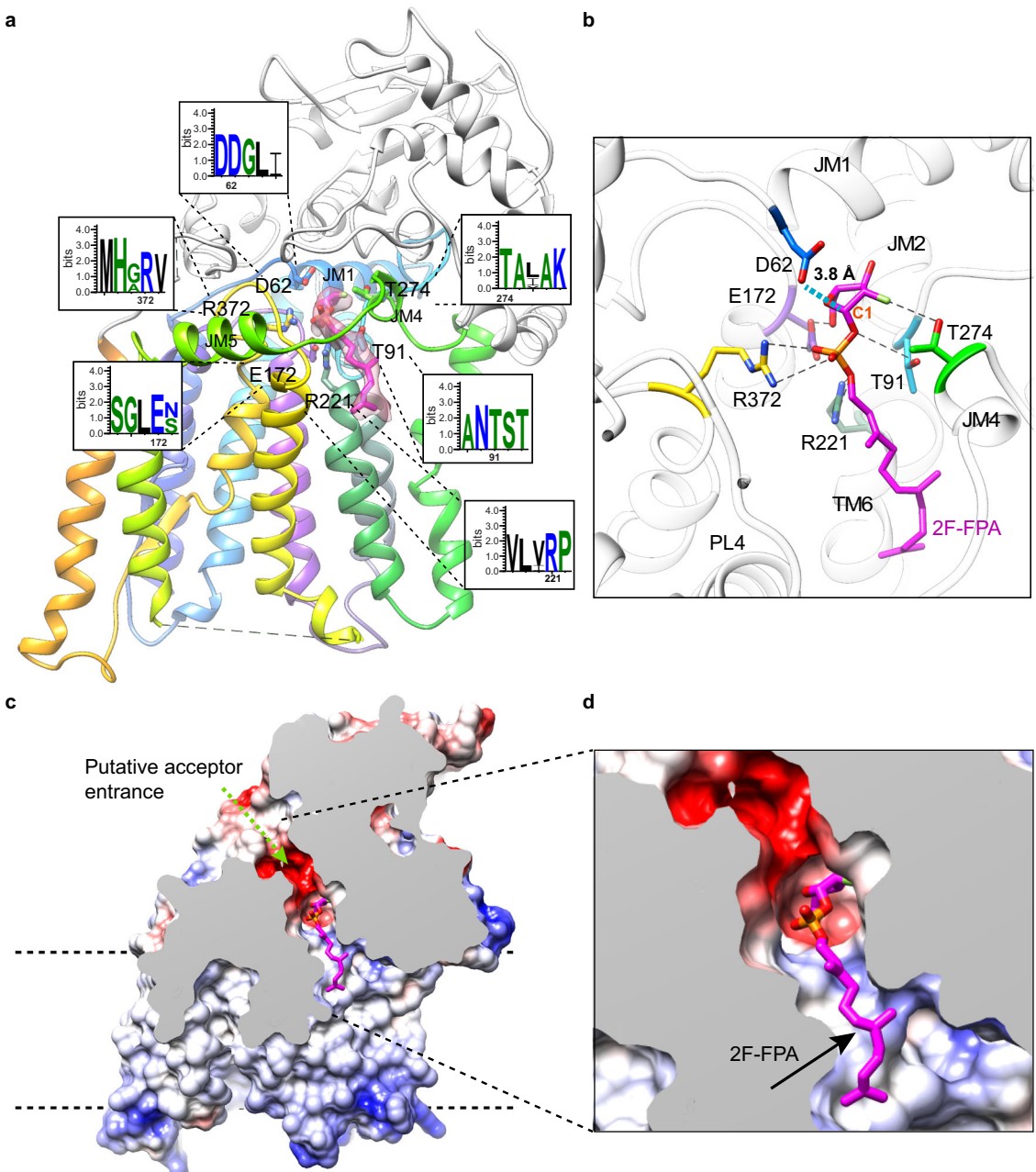

**Fig. 3 | Interaction between AftB and bound 2F-FPA. a** Structure of AftB complexed with 2F-FPA shown in ribbon. The TM domain of AftB is colored in rainbow as in Fig. 1d, while the PD is colored in white. 2F-FPA is shown in sticks and colored in magenta. *Cryo-EM density* of 2F-FPA colored with semi-semitransparent magenta. WebLogo plots[102] show the amino acids interacting with 2F-FPA. **b** Close-up view of the interaction between AftB and 2F-FPA. Polar contacts are indicated as black dashed lines. Residues involved in the interaction are shown as sticks. The distance between the catalytic residue D62 and C1 of 2F-FPA is indicated with blue dashed lines. **c**, **d** Electrostatic surface depiction of the AftB-2F-FPA complex, a partial cross-sectional perspective is applied to highlight the active site cavity with the bound donor substrate-analog 2F-FPA.

## The cryo-EM structure of 2F-FPA-bound AftB

To elucidate the molecular basis of substrate recognition and catalysis, a cryo-EM structure of AftB was determined in complex with 2-F-farnesyl phosphoarabinose (2F-FPA), a donor analog for mycobacterial arabinosyltransferases (AraTs)[57]. 2F-FPA substitutes the decaprenyl chain of DPA with a shorter, less hydrophobic farnesyl group, and incorporates a fluorine atom into the Araf ring to stabilize the glycosyl-phosphate bond (Supplementary Fig. 3d). This modification enhances its suitability for structural studies.

The structure was resolved to 3.4 Å using nanodisc-reconstituted AftB bound to 2F-FPA and Fab-B3 (Fig. 3 and Supplementary Fig. 4d–f). The entire protein was modeled except for 28 residues at the N-terminus, 11 residues at the C-terminus, a disordered cytoplasmic loop region (319–335) connecting TM helices 8 and 9, and a small periplasmic loop region (432–437) between TM helix 11 and periplasmic helix α1 (Supplementary Fig. 6). The Fab-B3 binding conformation is consistent with the apo structure (Supplementary Fig. 4d). The density for 2F-FPA was well resolved, revealing a partially curved conformation with the sugar ring projecting deep into the substrate cavity (Fig. 3a). The hydrophobic acyl chain extends along TM helix 6, while the phosphate moiety is coordinated by R221 and R372, with an additional hydrogen bond formed with T91, in the loop between JM helices 1 and 2 (Fig. 3b). T274 interacts with the fluorine atom on the arabinofuranose ring, and E172 forms a hydrogen bond with the C5 hydroxyl group of

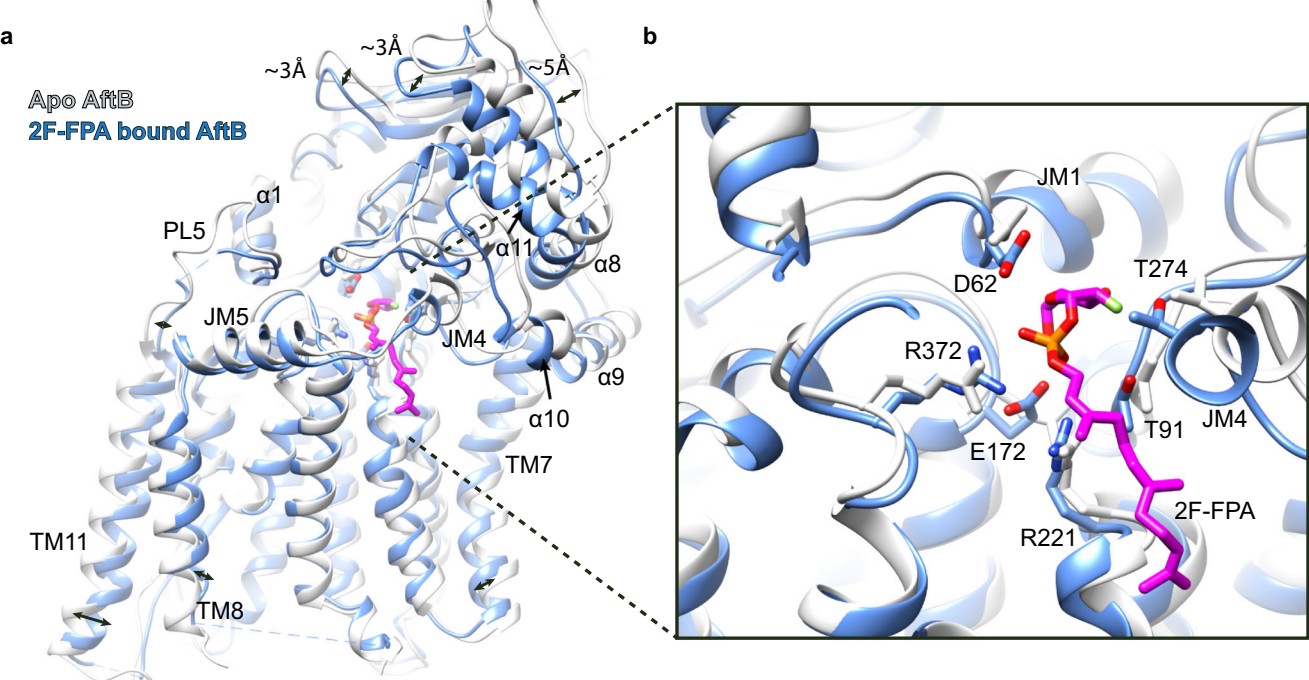

**Fig. 4 | Conformational rearrangement of AftB upon 2F-FPA binding.**
**a** Superimposition of apo (white) and 2F-FPA bound AftB (blue). PD undergoes a clamp-like movement while TM helices exhibit minor inward movement.

**b** Magnified view of the substrate pocket in the apo structure (white) and the 2F-FPA-bound structure (blue), superimposed, showing interacting residues shifted toward 2F-FPA.

the sugar (Fig. 3b). All five of the interacting residues noted above are conserved across mycobacterial homologs of AftB (Supplementary Fig. 8). Furthermore, mutants R221A, R221E, R372A, and R372E were catalytically inactive in functional assays (Supplementary Fig. 10), despite being expressed at levels comparable to wild-type *Mc*AftB (Supplementary Fig. 9), underscoring the important role of these residues in AftB activity. This structure provides key insights into substrate binding and catalysis, highlighting the conserved nature of substrate coordination in mycobacterial AraTs.

### Structural rearrangements upon substrate binding

Comparison of the apo and 2F-FPA-bound AftB structures shows conformational changes primarily within the PD. In the TM domain, TM helices 7 and 11 exhibit inward shifts, most pronounced in TM11. This is accompanied by a noticeable displacement of PL5, the loop connecting TM11 to α1 (Fig. 4). The shift in PL5 appears to act as a trigger for a conformational rearrangement in the PD, inducing a clamp-like movement of the domain towards the membrane (Fig. 4). This movement highlights the dynamic coupling between the TM domain and the PD during substrate binding.

Residues within the substrate-binding cavity—including D62, E172, R372, R221, T91, and T274—exhibit positional shifts towards the bound 2F-FPA, indicating a coordinated adaptation to facilitate substrate interaction (Fig. 4). These movements contribute to an inward pivoting of TM helices 1 and 7. The interplay between these structural rearrangements results in a more compact conformation of AftB.

### Insights into DPA and acceptor binding mode with docking, co-folding, and MD simulations

To advance our understanding of the interaction between the enzyme and its substrates, we used a combination of ligand docking, protein-ligand co-folding, and MD simulations. Ligand docking studies, coupled with Chai-1[58] and RoseTTAFold[59] analyses of the donor substrate propose that the native DPA substrate binds similarly to the 2F-FPA in the AftB-2F-FPA complex. Specifically, the polar head of DPA is

oriented towards D62 within the active site (Supplementary Fig. 11b, d). The phosphate group of DPA is coordinated by Arg372 and Arg221, along with interactions involving T91 and T243. This mirrors the coordination seen with 2F-FPA (Supplementary Fig. 11d).

To build on the Ara*f* binding site proposed by the earlier CG-MD simulations, we used a combination of ligand docking, with Autodock Vina[60], and protein-ligand co-folding, with both RoseTTAFold All-Atom[59] and Chai-1[58], to propose how the terminal-Ara*f*$_4$ and the subsequent products interact with AftB. The docking model suggests that two conserved residues, F448 and Y449, from the first periplasmic α1, are positioned near the branch point of terminal-Araf4. Y449 may form hydrogen bonds with the substrate, while F448 could contribute to substrate coordination through non-polar interactions (Supplementary Fig. 11d). Additional coordinating residues include T524 in the loop between periplasmic β3 and α4, D541 in the loop connecting periplasmic α4 to β4, and R560 and H563 in the loops linking β4 to α5 and α5 to α6 (Supplementary Fig. 11d). These residues appear to help position terminal-Ara*f*$_4$ close to the active site near DPA. The predicted importance of F448 and Y449 in acceptor substrate engagement is supported by the complete loss of enzymatic activity upon their mutation to alanine (Supplementary Fig. 10).

To further investigate the interaction of the substrates and products of this reaction with AftB, atomistic resolution MD simulations were performed. In these simulations, we used the native donor substrate DPA, its by-product decaprenyl-phosphate DP, and two polysaccharide mimics: terminal-Ara*f*$_4$, as described above (representing the acceptor substrate) (Supplementary Fig. 11a), and a "terminal-Ara*f*$_6$" (Supplementary Fig. 11c). The terminal-Ara*f*$_6$ represents the arabinofuranosylated product (Fig. 1a), characterized by the Ara*f*$_6$ motif at the end of the arabinan domain, including two additional α-(1 → 5)-Ara*f* residues from the linear backbone (Fig. 1a and Supplementary Fig. 11c).

We conducted MD simulations with either the substrates (DPA and terminal-Ara*f*$_4$) or the products (DP and terminal-Ara*f*$_6$) present. In addition, we modeled the state where the sidechain of D62 was

arabinofuranosylated, representing an intermediate state. In all simulation set-ups, the terminal-Ara$f_4$ and terminal-Ara$f_6$ dissociated from their bound poses (Supplementary Fig. 12), likely due to the lack of extended repeating sugar units and lipid membrane anchors. Conversely, the lipid donor (DPA), the sidechain of the arabinofuranosylated D62, and the by-product (DP) remained stably docked within the binding pocket (Supplementary Fig. 12). The addition of either substrates (DPA with terminal-Ara$f_4$) or products (DP and terminal-Ara$f_6$) had no significant effect on the overall dynamics of the protein compared to the apo state (Supplementary Fig. 13a, b).

## Discussion

AG and LAM/AM are complex polysaccharides that constitute major components of the mycobacterial cell envelope. We have determined the structures of AftB, an AraT responsible for adding terminal-Ara$f$ residues to both AG and LAM/AM, using cryo-EM in both its apo and 2F-FPA-bound state at resolutions of 2.9 Å and 3.4 Å, respectively.

These structures reveal that AftB belongs to the GT-C$_A$ subclass of GT-C glycosyltransferases, characterized by a conserved structural core of seven TM helices. Notably, an irregular tube-shaped cavity is observed, spanning the two proposed substrate-binding sites. The structural characteristics of the cavity, combined with experimental evidence, provide insights into substrate binding and enable us to propose a mechanism of action for AftB.

### Catalytic mechanism of AftB

Glycosyltransferases are classified as inverting or retaining based on the stereochemical outcome of the glycosyl transfer[51,61]. The former inverts the configuration of the anomeric carbon of the donor substrate in the resulting product[48,62], while the latter maintains the configuration[48,62,63]. AftB catalyzes the transfer of D-Ara$f$ from the donor DPA to the acceptor, forming a β-(1 → 2) glycosidic bond[35,36,64] (Fig. 1a). This reaction preserves the β-configuration of the anomeric carbon, identifying AftB as a retaining glycosyltransferase[28,35,36,64].

The catalytic mechanism of retaining glycosyltransferases is a subject of debate, with two main mechanisms proposed: the front-face S$_N$i (substitution nucleophilic internal) mechanism[62,63] or a double-displacement mechanism[51,62,65]. While most structurally characterized retaining glycosyltransferases lack a suitably positioned catalytic nucleophile, favoring the S$_N$i mechanism[62,63,65], recent structural and biochemical studies of [66,67] and KpsC[68], two β-Kdo transferases, have provided clear evidence for double-displacement mechanisms. These findings suggest this catalytic strategy may be more widespread among retaining glycosyltransferases than previously.

Our structural analysis reveals that D62, located within the conserved DD motif, is positioned ~3.8 Å away from the anomeric carbon (C1) of 2F-FPA, making it well-placed to function as a catalytic nucleophile. This distance is identical to that between the analogous amino acid and the donor anomeric center in the donor-bound complex of WbbB[61,66]. In addition, structural and computational studies on retaining glycoside hydrolases, which proceed via a double-displacement mechanism, have shown that the analogous nucleophile is typically positioned at distances of 3.2–3.9 Å from the anomeric center[69–72], consistent with our observations in the 2F-FPA bound structure. Supporting the critical role of D62, mutating it to alanine abolishes enzymatic activity. Furthermore, MD simulations indicate that the p$K_a$ of D62 decreases substantially in the presence of substrates or products (Supplementary Fig. 13c). This effect is not observed for other substrate-interacting residues (Supplementary Fig. 13c), which further emphasizes the enzymatic importance of D62. Based on these findings, we propose that at physiological pH, D62 would predominantly be in its deprotonated form, ready to perform a nucleophilic attack on the anomeric carbon (C1) of the DPA substrate.

Similar to what we observed in our 2F-FPA-bound structure, the MD contact analysis (Supplementary Fig. 12d) revealed that R221 forms

significant interactions with DPA, and both R221 and R372 show strong contacts with DP during simulations. Unlike D62, the p$K_a$ values of R221 and R372 remain relatively stable across different simulation conditions (Supplementary Fig. 13c), suggesting these residues maintain their protonated, positively charged states. Mutations of these conserved residues lead to a complete loss of AftB function (Supplementary Fig. 10), further underscoring their essential roles in the catalytic process.

Given these insights and AftB's retaining nature, we propose a double-displacement mechanism. The catalytic cycle begins with D62 initiating nucleophilic attack on the anomeric carbon of DPA (Fig. 5b). This results in the formation of a covalent α-Ara$f$–enzyme intermediate. This step inverts the stereochemistry of the Ara$f$ residue to be transferred. Concurrently, R221 and R372 stabilize the departing phosphate group of DPA which then deprotonates the C2 hydroxyl group (or polarizes the O–H bond) of the acceptor substrate (Fig. 5b), priming it for a nucleophilic attack.

The second step involves the C2 hydroxyl group of the acceptor attacking the anomeric carbon of the covalent α-Ara$f$–enzyme intermediate, breaking the arabinose–enzyme bond and forming a new β-(1 → 2) glycosidic bond. This second nucleophilic attack restores the original β-configuration of the donor Ara$f$ moiety, consistent with the retaining nature of AftB (Fig. 5b).

Although attempts to capture a ternary structure with donor and acceptor substrates were unsuccessful, structural and functional evidence supports a double-displacement mechanism for AftB. This mechanism accommodates AftB's bulky polysaccharide substrates, which may present steric constraints incompatible with an S$_N$i mechanism.

AftB represents a structurally characterized example of a retaining GT-C fold enzyme. This structure provides insights into retaining GT-C fold enzymes, though the absence of other structural examples in this class currently precludes more extensive comparative analysis.

Finally, no metal ions were observed in the active site in our cryo-EM structures of AftB. This observation aligns with findings from several other GT-C$_A$ enzymes; for instance, studies on both AftA and ALG6 have indicated that their activities do not require metal ions[30,55]. Similarly, our recent study on PimE[73], a mycobacterial mannosyltransferase also belonging to the GT-C$_A$ superfamily, demonstrated that PimE is metal-independent[73]. Nonetheless, future studies are needed to definitively determine if metal ions play any role in AftB function.

### Structural insights into the function of AftB

The donor substrate of AftB, decaprenyl-phosphoryl-arabinose (DPA), is a lipid-linked arabinose (Ara$f$) anchored to the membrane via a decaprenyl-phosphate moiety. Structural analysis reveals that the lipid tail of DPA extends out of the catalytic cavity towards TM helices 6 and 7, embedding into the lipid bilayer, while orienting the reactive Ara$f$–phosphate moiety towards the catalytic core.

The acceptor substrates for AftB are the polysaccharides AG and LAM/AM polysaccharides, both anchored to the membrane via a galactan or mannan core, respectively. These polysaccharides share conserved motifs at the non-reducing end of their arabinan domains, which serve as the acceptor sites for AftB. In AG, AftB features branched motifs at its arabinan termini, where AftB generates the characteristic Ara$f_6$ structure required for mycolic acid attachment. In contrast, LAM/AM biosynthesis involves both branched and linear arabinan regions, with AftB generating a terminal β-linked Ara$f_6$ structure. as well as linear Ara$f_4$ structures (Supplementary Fig. 1a).

The irregular tube-shaped catalytic cavity of AftB features a relatively spacious periplasmic-facing opening. This appears to be adapted to accommodate both branched and linear terminal arabinan acceptor motifs, enabling AftB to complete the final steps of both AG and LAM/AM biosynthesis. Further structural studies of AftB in complex with its

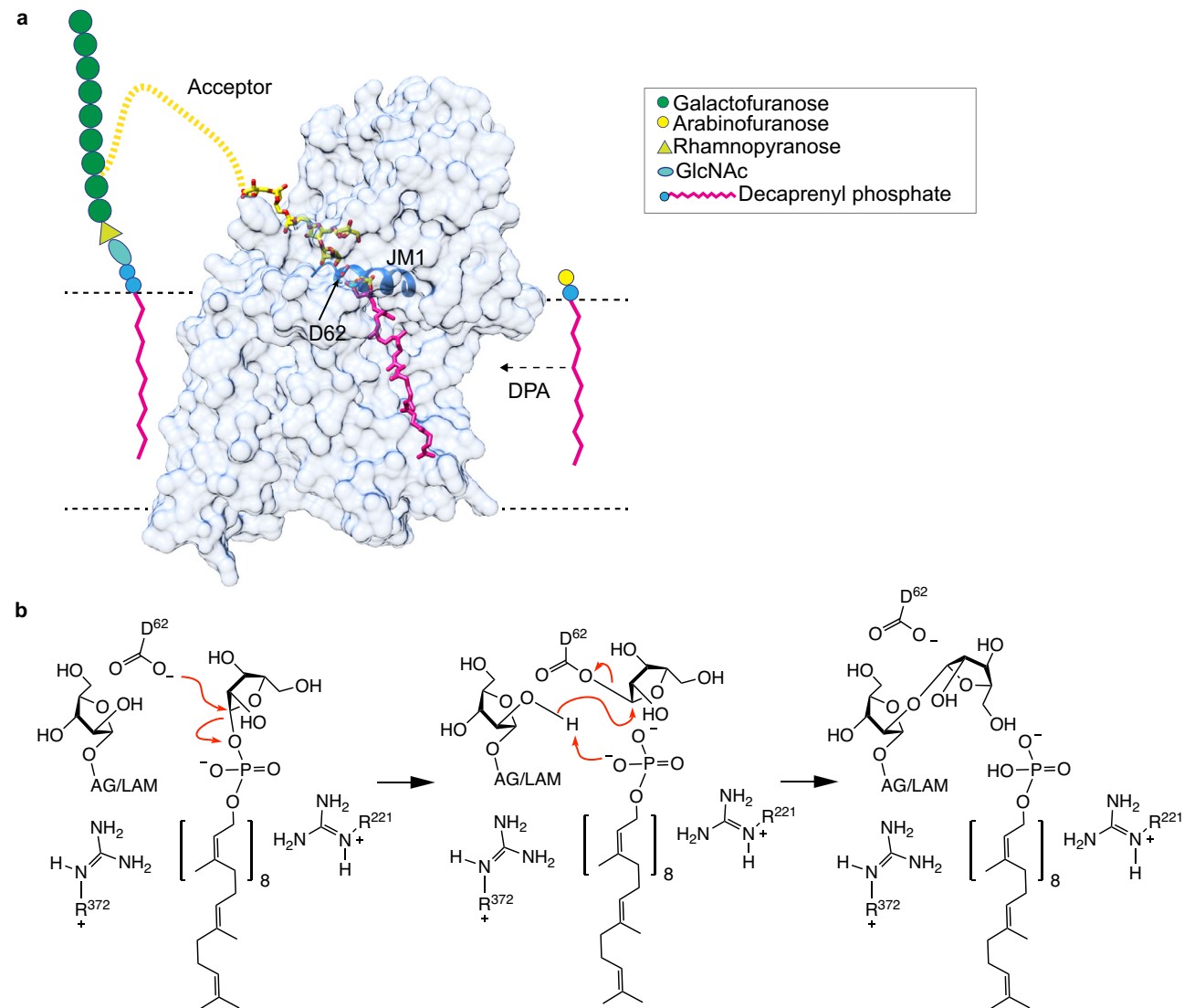

**Fig. 5 | Proposed mechanism for AftB-catalyzed retaining arabinosyl transfer.**
**a** Schematic representation of AftB's role in the biosynthesis of arabinogalactan. AftB directs the addition of arabinofuranosyl units from the donor molecule DPA onto the growing arabinan chain, maintaining the glycosidic bond stereochemistry. **b** Proposed catalytic model of AftB.

diverse acceptor substrates will enhance our understanding of substrate recognition and catalysis.

## Structural conservation across mycobacterial species

To evaluate the broader relevance of our *M. chubuense* AftB structure, structural comparisons were conducted with AlphaFold2 predictions of AftB orthologues from *M. tuberculosis*, the causative agent of tuberculosis, and *M. smegmatis*, a well-established model organism in mycobacterial research. The analysis revealed significant structural conservation across the species, with root-mean-square-deviation (RMSD) values of 2.8 Å (across 606 pairs) for *M. tuberculosis* AftB and 2.4 Å (across 617 pairs) for *M. smegmatis* AftB (Supplementary Fig. 14a, b). Sequence identity was also notably high, with 71% identity between *M. chubuense* and *M. tuberculosis* AftB, and 74% between *M. chubuense* and *M. smegmatis* AftB (Supplementary Fig. 14c). Key catalytic and substrate-binding residues identified in the *M. chubuense* AftB structure are identically positioned across the orthologues. This high degree of conservation strongly suggests that the structural and functional insights gained from the *M. chubuense* AftB structure can be extended to homologs from other mycobacterial species.

## Structural comparison with other mycobacterial arabinosyltransferases

The biosynthesis of mycobacterial arabinan involves a cascade of AraTs, each with distinct roles, with AftB adding the terminal β-(1 → 2) linkages via a retaining mechanism. Despite sharing a conserved GT-C$_A$ fold, these enzymes exhibit architectural variations that may be tailored to their specific functions. All these AraTs feature a conserved catalytic aspartate residue, but its position differs (Supplementary Fig. 3). In AftB, this residue is predicted to act as a nucleophile in a retaining double-displacement mechanism, whereas in other AraTs it serves as a general base to facilitate an inverting reaction.

AftB is characterized by an irregular, tube-shaped. The architecture of this cavity may be to accommodate AftB's dual role in binding both AG and LAM/AM acceptor substrates. AftA uses the galactan as its acceptor substrate, initiating arabinan synthesis, with structural features that appear to be adapted for selective galactan recognition[30] (Supplementary Fig. 3d). AftD presents a prominent cleft spanning its periplasmic and transmembrane domains, accompanied by an additional groove on its periplasmic face[74] (Supplementary Fig. 3f), in contrast to the more enclosed cavities observed in other AraTs.

For EmbA/B/C, the DPA donor substrate binds with its hydrophobic decaprenyl tail situated within a groove formed by transmembrane helices, while the sugar head group resides near the active site cavity[24,75] (Supplementary Fig. 3h). This binding mode is aligned with the observed DPA binding mode for AftB. However, the Emb enzymes exhibit a relatively constricted connection between their donor and acceptor binding sites.

AftD and EmbA/B/C possess a highly positively charged surface on the cytoplasmic side that facilitates ACP binding and a hydrophobic tunnel for accommodating the 4′-phosphopantetheine prosthetic group of ACP[24,74,75]. These motifs suggest an evolutionary adaptation for specialized functions, including co-functioning with ACP partner proteins[30]. In contrast, ACP is absent in the structures of AftB and AftA, although they both feature positively charged surfaces on their cytoplasm domains (Supplementary Fig. 4). These structural distinctions likely reflect the specialized catalytic roles and regulatory requirements of AraTs at different stages of arabinan biosynthesis. Further studies are required to elucidate the precise mechanisms underlying these variations.

## Therapeutic potential of targeting AftB

The mycobacterial cell wall, containing the key structural components AG and LAM/AM, is critical for pathogen resilience and virulence. Inhibiting the biosynthesis of these components, as demonstrated by the front-line anti-TB drug ethambutol targeting EmbB and EmbC, is a proven therapeutic strategy[21,76]. AftB catalyzes the final steps in AG and LAM/AM biosynthesis and plays an essential role in mycobacterial viability[35]. Depleting AftB has been shown to sensitize *M. tuberculosis* to rifampicin, a front-line TB drug targeting bacterial RNA polymerase, particularly under infection-relevant conditions[39]. This highlights AftB as a potential target for anti-TB drug development[35,36,39]. Additionally, conservation of AftB across pathogenic mycobacterial species, including non-tuberculous mycobacteria (NTM) such as *M. chubuense* (studied herein), *M. abscesses,* and *M. chelonae*[77–79] (Supplementary Fig. 8 and Supplementary Fig. 14), suggests its potential for broad-spectrum therapeutic applications. Targeting AftB could thus provide avenues to combat both drug-resistant and drug-sensitive mycobacterial pathogens[26,78].

Our structural studies provide insights into the substrate-binding cavity and active site architecture of AftB. Functional assays and computational analyses presented here provide further insight into how substrates interact with conserved residues within the cavity. This cavity could serve as a target for competitive inhibitors that either directly inhibit AftB or enhance the effectiveness of existing antibiotics like rifampicin. While future studies will be needed to fully explore the druggability of this cavity, our integrated characterization of AftB provides a foundation for further investigation of this essential enzyme as a therapeutic target[77–79].

## Methods

### Overexpression and purification of *Mycobacterium chubuense* AftB

Wild-type *Mycobacterium chubuense* AftB was cloned into pNY-COMPSC23 plasmid[80]. The procedure for design and synthesis of pNYCOMPSC23 plasmid was described in detail in a previous protocol[80]. The plasmids were transformed into BL21 (DE3) pLysS *E. coli* for protein expression. The following day, a single colony was selected to inoculate a starter culture consisting of 50 mL Terrific Broth (TB) medium (Fisher), supplemented with 50 µg/mL ampicillin and 35 µg/mL chloramphenicol. The starter culture was grown overnight at 37 °C with shaking at 220 rpm in an incubator shaker (New Brunswick Scientific). The next day, four flasks, each containing 1 L of TB medium with 50 µg/mL ampicillin and 35 µg/mL chloramphenicol, were inoculated with 10 mL of the starter culture and grown at 37 °C with shaking at 220 rpm until the OD600 reached 0.6–0.8. The

cultures were then cooled to 22 °C, and protein expression was induced by adding 0.5 mM IPTG for 16 h. Cells were harvested by centrifugation at $4000 \times g$ in H6000A/HBB6 rotor (Sorvall) for 30 min at 4 °C, and the pellet was resuspended with lysis buffer containing 20 mM HEPES pH 7.5, 200 mM NaCl, 20 mM $MgSO_4$, 10 mg/mL DNase I (Roche), 8 mg/mL RNase A (Roche), 1 mM tris(2-carboxyethyl) phosphine hydrochloride (TCEP), 1 mM PMSF, 1 tablet/1.5 L buffer EDTA-free cOmplete protease inhibitor cocktail (Roche). Cells in lysis buffer were lysed by passing through a chilled Emulsiflex C3 homogenizer (Avestin) three times. The lysate was centrifuged at $3000 \times g$ in a Centrifuge 5810 R (Eppendorf) at 4 °C for 5 min to remove cell debris and non-lysed cells. To isolate the cell membrane, the supernatant was ultracentrifuged in a Type 45 Ti Rotor (Beckman Coulter) at $185,600 \times g$ for 1 h. For 4 L of culture, the yield was ~4 g of membrane pellet, which was resuspended in the lysis buffer up to 80 mL and homogenized using a handheld glass homogenizer (Kontes) on ice. The membrane fraction was stored at −80 °C until further use. The membrane fraction was thawed and solubilized by adding *n*-dodecyl-β-D-maltopyranoside (DDM) to a final concentration of 1% (w/v) and was rotated gently at 4 °C for 2 h. Insoluble material was removed by ultracentrifugation at $185,600 \times g$ in a Type 45 Ti Rotor at 4 °C for 40 min. The supernatant was added to preequilibrated $Ni^{2+}$-NTA resin (QIAGEN) in the presence of 40 mM imidazole and incubated with gentle rotation at 4 °C for 2 h.

The resin was then washed with 10 column volumes of wash buffer containing 20 mM HEPES pH 7.5, 200 mM NaCl, 65 mM imidazole, 0.1% DDM and eluted with 3 column volumes of elution buffer containing 20 mM HEPES pH 7.5, 200 mM NaCl, 300 mM imidazole, 0.03% DDM. The eluted protein was exchanged into a buffer containing 20 mM HEPES pH 7.5, 200 mM NaCl, 0.03% DDM using a PD-10 desalting column (GE).

To characterize the expression and folding properties of *Mc*AftB mutants relative to wild-type protein, both wild-type and mutant *Mc*AftB were expressed from 250 mL cultures following identical procedures. All purification steps remained the same as described above. After buffer exchange using the PD-10 desalting column, both wild-type and mutant proteins were analyzed by size-exclusion chromatography using a Superose™ 6 Increase 5/150 GL column to assess their oligomeric state and folding properties.

### Preparation of mycobacterial membranes for arabinosyltransferase assay

Wild-type *M. smegmatis* mc²155 cells were grown in 50 mL of Luria–Bertani (LB) broth for 72 h at 37 °C and 200 rpm. An Erlenmeyer flask with 2 L of LB broth was inoculated with 10 mL of the saturated starter culture and incubated for 48 h at 37 °C and 130 rpm. Cells were harvested by centrifugation at $2600 \times g$ for 15 min at 4 °C, resuspended in ice-cold buffer A [50 mM MOPS (pH 7.9), 5 mM 2-mercaptoethanol and 10 mM $MgCl_2$], and disrupted by passing the suspension through a high-pressure homogenizer at 20,000 psi. The lysate was centrifuged at $27,000 \times g$ for 30 min at 4 °C. The cell wall pellet was removed, and the supernatant was recentrifuged at $100,000 \times g$ for 2 h at 4 °C. The supernatant was discarded, and the pellet of enzymatically active membrane was gently resuspended in 1 mL of buffer A with gentle homogenization using a Dounce homogenizer. The protein concentration of the membrane fractions was determined using the Pierce BCA protein assay kit and was typically between 15 and 20 mg/mL.

### Arabinosyltransferase assay

The arabinosyltransferase activity was assessed using wild-type *M. smegmatis* membranes or *E. coli* membranes expressing *Mc*AftB as described above. Untransformed *E. coli* membranes were used as a negative control. A typical reaction mixture contained FPA (1 mM), synthetic acceptor (0.2 mM), ATP (1 mM), DMSO (2% v/v), buffer A [50 mM MOPS (pH 7.9), 5 mM 2-mercaptoethanol and 10 mM $MgCl_2$]

and the respective membranes (2 mg) in a total volume of 200 μL. The reaction mixture was incubated at 37 °C overnight and then terminated by adding 200 μL of ethanol. The resulting mixture was centrifuged at 20,000 × g for 15 min, and the supernatant was loaded onto a preequilibrated strong anion exchanger (SAX) cartridge. The cartridge was eluted with 5 mL of 50% ethanol–water. The eluate was evaporated to dryness, and the dried enzymatic products were treated with $Ac_2O$ (100 μL) in pyridine (100 μL). The reaction mixture was stirred overnight at room temperature before being concentrated and then partitioned between the two phases (1:1) of chloroform and water. The organic layer was concentrated before being analyzed by MALDI mass spectrometry. Samples for MALDI analysis were prepared by 10-fold dilution in methanol or water. The MALDI matrix was prepared by dissolving 2,5-dihydroxybenzoic acid (DHB, 10 mg) in 1 mL of 50% ACN. Next, spotting 1 μL of DHB matrix and then 1 μL of the diluted sample onto the MALDI target plate was followed by mixing by gentle pipetting on the plate. Samples were allowed to dry completely (15–20 min) before analysis. MALDI-MS measurements were carried out on a Bruker ULTRAFLEXTREME instrument in reflector mode. The mass range investigated was 600–3500 m/z in positive ion mode using 500 Da matrix suppression. The laser was operated at a frequency of 50 Hz using the "large" predefined shot pattern and 4000 laser shots were collected per spectrum. The following settings were used: Ion Source 1: 20 kV, Ion Source 2: 17.75 kV, Lens: 7 kV, Reflector: 20.8 kV, Reflector 2: 10.95 kV. Data analysis was done using the following software: Compass for flexSeries 1.4, flexControl Version 3.4, and flexAnalysis Version 3.4.

## MS/MS on $Araf_5$ and $Araf_6$ products from enzymatic reactions

Enzymatic synthesized products were permethylated and analyzed by MALDI-MS/MS in reflector positive ion mode as described previously[42,43], but using instead an AB SCIEX MALDI TOF/TOF 5800 system. MS/MS spectra were acquired with a laser intensity at 5000 and 1 kV CID energy. Five MS/MS spectra (1000 laser shots/spectrum) were accumulated to yield the final spectra (total laser shots = 5000) that were assigned manually according to the fragmentation pattern established in previous analyses of permethylated arabinan oligomers[42,43].

## Nanodisc reconstitution of AftB

The protein was incorporated into lipid nanodiscs with a molar ratio 1:5:250 of membrane scaffold protein 1E3D1 (MSP1E3D1): (POPG) 1-palmitoyl-2-oleoyl-sn-glycero-3-phospho-(1′-rac-glycerol) (Avanti), and incubated for 2 h with gentle agitation at 4 °C. For the substrate-bound structure, 2F-FPA was added to the reconstitution mixture at a molar ratio of 1:3 (AftB: 2F-FPA). The POPG lipid was prepared by adding the solid extract to deionized water to a final concentration of 20 mM. The mix was put on ice and then gently sonicated with a tip sonicator (Fisher Scientific) to dissolve the lipids until the mixture became semitransparent.

To initiate nanodisc reconstitution, 100 mg Biobeads (Bio-Rad) per mL of protein solution was added to the mixture and incubated by gentle rotation at 4 °C overnight. Biobeads were removed the next day by filtering the reconstitution mixture through an Ultrafree centrifugal filter unit (Fisher) at 16,100 × g in a Centrifuge 5415 R (Eppendorf) at 4 °C for 1 min. The reconstitution mixture was rebound to $Ni^{2+}$-NTA resin in the presence of 25 mM imidazole for 2 h at 4 °C in order to remove free nanodisc. The resin was washed with 10 column volumes of wash buffer containing 20 mM HEPES pH 7.5, 200 mM NaCl, and 50 mM imidazole, followed by 4 column volumes of elution buffer containing 20 mM HEPES pH 7.5, 200 mM NaCl, and 300 mM Imidazole. The eluted protein was subsequently purified by size-exclusion chromatography (SEC) on a Superdex 200 Increase 10/300 GL column in buffer containing 20 mM HEPES pH 7.5, 200 mM NaCl.

## Cryo-EM sample preparation

Fractions containing AftB incorporated into nanodiscs were pooled and incubated with the Fab at 4 °C for 2 h in a 1:3 molar ratio of protein to Fab. The protein mixture was further purified using a Superdex 200 Increase 10/300 GL SEC column in SEC buffer containing 20 mM HEPES pH 7.5, 200 mM NaCl.

For the apo structure, peak fractions were pooled and concentrated to 7 mg/mL using a 50 kDa cut-off filter concentrator (Amicon). The sample was frozen using a Vitrobot (Thermo Fisher) by applying 3 μL to a plasma cleaned (Gatan Solarus) 0.6/1-mm holey gold grid (Quantifoil UltrAuFoil). After a 30 s incubation, the grids were blotted using 595 filter paper (Ted Pella, Inc) for 8 s before being immediately plunged into liquid ethane for vitrification. The plunger was operated at 4 °C with greater than 90% humidity to minimize evaporation and sample degradation. For the substrate-bound structure, peak fractions were pooled and concentrated to concentration of 5 mg/mL. The sample was incubated with 2F-FPA at a 1:2 molar ratio of protein to substrate for 1 h before being applied onto the grids. The grids were prepared in the same way as for the apo sample, but with a blotting time of 6 s.

## Data collection

For the apo structure, images were recorded at the Columbia University Cryo-Electron Microscopy Center on a Titan Krios electron microscope (FEI), equipped with an energy filter and a K3 direct electron detection filter camera (Gatan K3-BioQuantum) using a 0.87 Å pixel size. An energy filter slit width of 20 eV was used during the collection and was aligned automatically every hour using the Leginon software package[81]. Data collection was performed using a dose of around 58 $e^-$ per $Å^2$ across 50 frames (50 ms per frame) at a dose rate of ~16 $e^-$ per pixel per second, using a set defocus range of −1.2 μm to −2.2 μm. A total of 7164 micrographs were collected over a single 2-day session.

For the substrate-bound structure, images were recorded at the New York Structural Biology Center (NYSBC) on a Titan Krios (FEI; NYSBC Krios 2) operating at 300 kV equipped with a spherical aberration corrector, an energy filter (Gatan GIF Quantum), and a post-GIF K2 Summit direct electron detector, using a 0.8460 Å pixel size. An energy filter slit width of 20 eV was used during the collection and was aligned automatically every hour using the Leginon software package[81]. Data collection was performed using a dose of around 50.3 $e^-$ per $Å^2$ across 24 frames (50 ms per frame) at a dose rate of ~41 $e^-$ per pixel per second, using a set defocus range of −0.8 μm to −2.5 μm. A total of 16,290 micrographs were collected over a single session of 3 days and 20 h (92 h total).

## Identification of *Mc*AftB-specific Fab using phage display

AftB was reconstituted into chemically biotinylated MSP1E3D1 as previously described[44,82]. Selection for Fabs was performed starting with Fab Library E[45,83]. Targets and the library were first diluted in selection buffer (20 mM HEPES, pH 7.4, 150 mM NaCl, and 1% BSA). Five rounds of sorting were performed using a protocol adapted from published protocols[84,85]. In the first round, bio-panning was performed manually using 400 nM of AftB, which was first immobilized onto magnetic beads and washed three times with selection buffer. The library was incubated for 1 h with the immobilized target, beads were subsequently washed three times with selection buffer, and then beads were used to directly infect log-phase *E. coli* XL-1 Blue cells. Phage were amplified overnight in 2XYT media supplemented with ampicillin (100 μg/mL) and M13-K07 helper phage ($10^9$ pfu/mL). To increase the stringency of selection pressure, four additional rounds of sorting were performed by stepwise reduction of the target concentration: 200 nM in the 2nd round, 100 nM in the 3rd round, and 50 nM in the 4th and 5th rounds. These rounds were performed semi-automatically using a KingFisher magnetic beads handler (Thermo Fisher Scientific).

For each round, the amplified phage population from each preceding round was used as the input pool. Additionally, amplified phage was precleared prior to each round using 100 μL of streptavidin paramagnetic particles, and 2.0 μM of empty MSP1E3D1 nanodiscs were used throughout the selection as competitors in solution. For rounds 2–5, prior to infection of log-phage cells, bound phage particles were eluted from streptavidin beads by 15 min incubation with 1% Fos-choline-12 (Anatrace).

## Single-point phage ELISA to validate Fab binding to *AftB*
96-well plates (Nunc) were coated with 2 μg/mL Neutravidin and blocked with selection buffer. Colonies of *E. coli* XL-1 Blue cells harboring phagemids from the 4th and 5th rounds were used to inoculate 400 μL 2XYT media supplemented with 100 μg/mL ampicillin and $10^9$ pfu/mL M13-KO7 helper phage, and phage were subsequently amplified overnight in 96-well deep blocks with shaking at 280 rpm. Amplifications were cleared of cells with a centrifuge step and then diluted 10-fold into ELISA (selection buffer with 2% BSA). All phage were tested against wells with immobilized biotinylated MSP1E3D1-reconstituted AftB (30 nM), empty biotinylated-MSP1E3D1 nanodiscs (50 nM), or buffer alone to determine specific target binding. Phage ELISA was subsequently performed as previously described[82,84] where the amount of bound phage was detected by colorimetric assay using an anti-M13 HRP-conjugated monoclonal antibody (GE Healthcare; Code: 27-9421-01; 1:5000 dilution). Binders with high target and low non-specific signal were chosen for subsequent experiments.

## Fab cloning, expression, and purification
Specific binders based on phage ELISA results were sequenced at the University of Chicago Comprehensive Cancer Center DNA Sequencing facility and unique clones were then sub-cloned into the Fab expression vector RH2.2 (kind gift of S. Sidhu) using the In-Fusion Cloning kit (Takara). Successful cloning was verified by DNA sequencing. Fabs were then expressed and purified as previously described[82]. Following purification, Fab samples were verified for purity by SDS-PAGE and subsequently dialyzed overnight in 20 mM HEPES, pH 7.4, 150 mM NaCl.

## Assessment of Fab binding affinity to AftB
To measure the apparent binding affinity, multi-point ELISAs using each purified Fab were performed in triplicate. Briefly, AftB (30 nM) or empty biotinylated-MSP1E3D1 nanodiscs (50 nM) were immobilized onto 96-well plates coated with Neutravidin (2 μg/mL). Fabs were diluted serially 3-fold into ELISA buffer using a starting concentration of 3 μM, and each dilution series was tested for binding to wells containing either AftB, empty nanodiscs, or no target at all. The Fab ELISA was subsequently performed as previously described[84], where the amount of bound Fab was measured by a colorimetric assay using an HRP-conjugated anti-Fab monoclonal antibody (Jackson ImmunoResearch). Measured $A_{450}$ values were plotted against the log Fab concentration, and $EC_{50}$ values were determined in GraphPad Prism version 8.4.3 using a variable slope model assuming a sigmoidal dose response.

## Single-particle cryo-EM data processing and map refinement
All data sets were corrected for beam-induced motion with Patch Motion Correction implemented in cryoSPARC v.2.15[86] and the contrast transfer function (CTF) was estimated with Patch CTF. For the apo structure, 5.03 million particles were automatically picked using a blob-picker job and subjected to multiple rounds of 2D classification. Representative 2D classes of 118,434 particles clearly showing the Fab-bound complex in the side views were selected as input for Topaz training. The resulting model was used to pick particles using Topaz Extract. Initially, 2,790,710 particles were extracted with a box size of 320 pixels and binned four times. Multiple rounds of 2D classification were performed to clean up the particles using a batchsize per class of

400 and "Force Max over poses/shifts" turned off with 40 online-EM iterations. Classes of total 335,357 particles which display clear features of a Fab-bound nanodisc-embedded membrane protein were reextracted using a 360-pixel box size without binning. Ab initio reconstruction was performed in cryoSPARC v.2.15 using three classes and a class similarity parameter of 0.1. One good class comprised of 241,095 particles was subjected to heterogeneous refinement and a final class comprising 192,312 particles was subjected to Nonuniform refinement and yielded a reconstruction with a resolution of 3.0 Å (FSC = 0.143). A subsequent local refinement was performed using a mask covering AftB and the variable region of the Fab and resulted in a density map at 2.85 Å resolution.

For the substrate-bound structure, 5.29 million particles were automatically picked using a blob-picker job and subjected to multiple rounds of 2D classification. Representative 2D classes of 10,186 particles clearly showing the Fab-bound complex in the side views were selected as input for Topaz training. The resulting model was used to pick particles using Topaz Extract. Initially, 2.01 million particles were extracted in a box of 320 pixels and Fourier cropped to 80 pixels for initial cleanup. Multiple rounds of 2D classification were performed to clean up the particles using a batchsize per class of 400 and "Force Max over poses/shifts" turned off with 40 online-EM iterations. Classes of total 279,138 particles which display clear features of a Fab-bound nanodisc-embedded membrane protein were reextracted using a 360-pixel box size without binning. Ab initio reconstruction was performed in cryoSPARC v.2.15 using two classes and a class similarity parameter of 0.1. One good class comprised of 183,657 particles was subjected to heterogeneous refinement and a final class comprised of 146,894 particles was subjected to Nonuniform refinement and yielded a reconstruction with a resolution of 3.8 Å (FSC = 0.143). A subsequent local refinement was performed using a mask covering AftB and the variable region of the Fab and resulted in a density map at 3.40 Å resolution.

## Model building
All model building was performed in Coot[87]. The apo AftB model was constructed de novo from the globally sharpened 2.85 Å map using the Phenix Map to Model tool[88]. Manual segment joining and residue assignment were guided by Xtalpred's secondary structure prediction[89]. Ramachandran outliers were fixed manually in Coot before the structure was refined with Phenix real space[88] refine with secondary structure and Ramachandran restraints

For substrate-bound AftB, the ligand files were generated in Coot, and similarly to the apo structure, manual model building, and refinement were conducted iteratively with Coot[87] and Phenix[39]. The structural quality of both apo and substrate-bound forms was assessed using Molprobity[90]. For most of the Fab region, with the exception of the binding interface, modeling was based on an existing high-resolution structure (PDB ID 5UCB [https://doi.org/10.2210/pdb5UCB/pdb]).

## Molecular dynamics simulations
The CG parameters for DPA and terminal-Ar*af*₄ were generated based on previously published lipid[91] and Martini 3 parameters[92–94]. The values for bond lengths and angles were generated from AT simulation data, where the molecules of interest were parameterized using CGenFF[95], and PyCGTOOL[96] was used to get exact values.

For CG simulations, the apo structure (with or without the Fab bound) was converted to the Martini 3 forcefield using martinize2[97] including a 1000 kJ mol$^{-1}$ nm$^{-2}$ elastic network and default protonation states. The structure was then embedded in a PE:PG (80:20) membrane using the insane script[98]. Each simulation box was then solvated and neutralized with 150 M NaCl. For DPA simulations, 2% DPA was added to the upper leaflet of the membrane. For the simulations including the terminal-Ar*af*₄, two copies of coarse-grained substrate were added to

the solvent in a random orientation and positioning. In each simulation, the system was comprised of ~40,000 particles (~30,000 of these being water molecules) in a $14 \times 18 \times 18$ nm$^3$ box. Systems were energy minimized using the steepest descents method. Production simulations used a timestep of 20 fs, and five simulations of 3 μs were performed for both DPA and terminal-Ara$f_4$ containing systems. The Parrinello-Rahman barostat[99] was set at 1 bar and the velocity-rescaling thermostat[100] was used at 310 K. The Reaction-Field algorithm was used for electrostatic interactions with a cut-off of 1.1 nm. A single cut-off of 1.1 nm was used for van der Waals interactions. All simulations were performed using GROMACS 2021.4[101,102].

Coordinates for the atomistic systems were generated from the end snapshots of the CG simulations using CG2AT2[103]. Coordinates for both the substrates and products were built using the densities from the ligand-bound cryo-EM structure. For the substrate state the bound DPA analog was modified to DPA and energy minimized. RoseTTAFold All-atom[59] was used to fold and dock both substrates and products for comparison. Atomistic molecular simulations were performed with the CHARMM36m forcefield[104], with parameters for the terminal-Ara$f_4$ and terminal-Ara$f_6$ (substrates and products respectively) created using CHARMM-GUI[105]. Parameters for the polyprenyl phosphate and DPA were created using CHARMM-GUI and combined with previous parameters from the past studies[106,107]. Energy minimizations of the systems were performed using the steepest descents method. A timestep of 2 fs was used for production simulations, with three independent simulations of 500 ns for substrate, product, Fab bound, and apo states of AftB. The C-rescale barostat[108] was set at 1 bar and the velocity-rescaling thermostat[100] was used at 310 K. The PME algorithm was used for electrostatic interactions with a cut-off of 1.2 nm. A single cut-off of 1.2 nm was used for van der Waals interactions. All simulations were performed using GROMACS 2021.4[101,102]. In each simulation, the system was comprised of ~222,000 particles (~50,000 of these being water molecules) in a $13 \times 15 \times 10$ nm$^3$ box.

For analysis, PyLipID[109] was used to measure interactions between the substrates and the protein. PLUMED[110] was used in conjunction with Matplotlib[111] to generate the density plots. The p$K_a$ of the residues were measured over the course of the simulations with PROPKA3[112] and propkatraj (version 1.1.0). Visualization of the duration was performed with Visual Molecular Dynamics (VMD)[113].

### Reporting summary

Further information on research design is available in the Nature Portfolio Reporting Summary linked to this article.

## Data availability

The MD data can be found at: https://doi.org/10.5281/zenodo.14916580. The cryo-EM maps and atomic coordinates have been deposited through the wwPDB OneDep system, with map accession numbers EMD-46976 (apo AftB) and EMD-46978 (2F-FPA-bound AftB). The atomic coordinate accession numbers are 9DLF (apo AftB) and 9DLH (2F-FPA-bound AftB). The PDB code of the previously published structure used in this study is 5UCB. Source data are provided as a Source Data file. Source data are provided with this paper.

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

## Acknowledgements

This work was supported by R35GM132120 (to F.M.). P.J.S. acknowledges R01AI174416 (PI: M. Stephen Trent), Wellcome (208361/Z/17/Z), MRC, BBSRC (BB/P01948X/1), and the Howard Dalton Centre for funding. P.J.S. and C.M.B. acknowledge Sulis at HPC Midlands+, which was funded by the EPSRC on grant EP/T022108/1, and the University of Warwick Scientific Computing Research Technology Platform for computational access. C.M.B. was supported by an MRC studentship (MR/N014294/1). T.L.L. thanks Academia Sinica intramural funding and the Canadian Glycomics Network (SD-1) for support. K.H.K. thanks Academia Sinica intramural funding. This project made use of time on ARCHER2 and JADE2 granted via the UK High-End Computing Consortium for Biomolecular Simulation, HECBioSim (http://www.hecbiosim.ac.uk), supported by EPSRC (grant no. EP/R029407/1). M.A. acknowledges support by FCT—Fundação para a Ciência e a Tecnologia, I.P., through PTDC/BIA-BQM/4056/2020, MOSTMICRO-ITQB R&D Unit (DOI 10.54499/UIDB/04612/2020; DOI 10.54499/UIDP/04612/2020) and LS4FUTURE Associated Laboratory (DOI 10.54499/LA/P/0087/2020); and EU H2020 grant No 823780. Mass spectrometry data for products from the functional assay were acquired at the Medicinal Chemistry and Analytical Core Facilities in the Biomedical Translation Research Center located at National Biotechnology Research Park, supported by Academia Sinica Core Facility and Innovative Instrument (AS-NBRPCF-111-201). A.A.K. acknowledges support from NIH grant R01GM117372.

## Author contributions

Y.L. and B.K. conducted the initial high-throughput screening and J.R. contributed to these early stages of the project. Y.L. performed molecular cloning, protein expression, purification, cryo-EM sample preparation, data processing, model building, and structural refinement. Y.C.S. and Y.J.W. conducted the AftB functional assay. P.S.T. and N.H.D. synthesized FPA for the functional assay. C.R.H. and H.Y.C. synthesized the Ara$f_4$ acceptor for the functional assay. Contributions related to the functional assay were made under the guidance of T.L.L. The MS/MS experiments used to support the structure of the products from the functional assays were carried out by S.Y.G. under the supervision of K.H.K. C.M.B. and P.J.S. carried out all molecular dynamics simulations and docking studies. S.E. and P.T. identified, characterized, and purified the Fabs under the supervision of A.A.K. Y.L. wrote the manuscript with critical input from R.N., F.M., P.J.S., C.M.B., M.A., T.L.L., and S.E. T.L.L., F.M., R.N., and P.J.S supervised the project. All authors reviewed and approved the final version of the manuscript.

## Competing interests

The authors declare no competing interests.

## Additional information

¹Department of Physiology and Cellular Biophysics, Columbia University Irving Medical Center, New York, NY, USA. ²School of Life Sciences & Department of Chemistry, University of Warwick, Coventry, UK. ³Groningen Biomolecular Sciences and Biotechnology Institute and Zernike Institute for Advanced Materials, University of Groningen, Groningen, The Netherlands. ⁴Department of Biochemistry and Molecular Biophysics, University of Chicago, Chicago, IL, USA. ⁵Institute of Biological Chemistry, Academia Sinica, Taipei, Taiwan, ROC. ⁶Department of Chemistry, University of Alberta, Edmonton, AB, Canada. ⁷Chemical Biology and Molecular Biophysics, Taiwan International Graduate Program, Academia Sinica, Taipei, Taiwan, ROC. ⁸Department of Chemistry, National Tsing Hua University, Hsinchu, Taiwan, ROC. ⁹Instituto de Tecnologia Química e Biológica António Xavier, Universidade Nova de Lisboa (ITQB-UNL), Oeiras, Portugal. ¹⁰Institute of Biochemical Sciences, National Taiwan University, Taipei, Taiwan, ROC. ¹¹Department of Radiation Oncology, Weill Cornell Medicine, New York, NY, USA. ✉e-mail: tlowary@as.edu.tw; Phillip.Stansfeld@warwick.ac.uk; rin7007@med.cornell.edu; fm123@cumc.columbia.edu

