## [Transparent Peer Review file · Nature Communications]

Structural insights into terminal arabinosylation biosynthesis of the mycobacterial cell wall arabinan

Corresponding Author: Professor Filippo Mancia

Version 0:

Reviewer comments:

Reviewer #1

(Remarks to the Author)

The manuscript by Liu et al. describes insights into the terminal arabinosylation biosynthesis of the mycobacterial cell wall. It follows on from earlier publications which have characterized the detailed structural biology (by Cryo-EM and X-ray crystallography) and biochemistry of the related *Mycobacterium tuberculosis* arabinosyltransferases EmbA, B and C proteins reported by Zhang et al. in *Science* (2020) 368(6496), 1211, and *Protein Science* (2020) 11(7), 505, AftD by Tan et al. in *Molecular Cell* (2020) 78(4), 683, and AftA by Gong et al. in *PNAS* (2023) 120(23), e2302858120.

At first glance, it is unclear how the current manuscript on AftB, other than being a retaining β -arabinosyltransferase, whereas the others are inverting α -arabinosyltransferases, based on the donor decaprenolphosphoarabinose (DPA), provides a conceptual advance to this class of GT-C enzymes, and in particular due to the lack of biochemical data linked to the analysis of purified mutated AftB proteins, which would unequivocally demonstrate activity and mechanistic insights into AftB.

Major Points

It was notable that two key errors describing areas relevant to the present literature have been made in the current manuscript relevant to α -arabinosyltransferases, which must be corrected. In addition, the biochemical data is insufficient evidence in confirming that AftB is an β -arabinosyltransferase, although it is known to be in the original cited literature. Furthermore, the authors should extend their analyses to include mutated purified AftB proteins to probe the mechanism and function of AftB.

a) Line 74 – This is followed by the action of enzymes EmbA and EmbB, which extend the arabinan chain through α -(1-5) linkages 31 (Fig. 1a).

This is factually incorrect. If the authors were to re-read citation (31, Escuyer et al. [2001] *J. Biol. Chem.* 276(52), 48854) as listed in the references list, it was suggestive that EmbA and EmbB were hypothesised as α -(1-3) linkages involved in the terminal stages of arabinan synthesis and NOT α -(1-5) linkages. The definitive evidence really came from the recent Zhang et al. publication in *Science* (2020) 368(6496), 1211, which clearly demonstrated that EmbA/B formed a novel protein complex and the activity was indeed an α -(1-3) linkage at the terminal stages of arabinan synthesis. This sentence needs to be corrected based on findings in the citation of Zhang et al. as the present citation is outdated as it refers to literature which was speculative at the time, and the correct citation of Zhang et al. used at the appropriate places in the manuscript.

b) AftD is thought to have a similar function to AftC, yet its precise role remains not fully elucidated 34, 35.

This is again factually incorrect. The authors need to refer to the citation of Alderwick et al. (2018) *Cell Surf.* 1, 2, which provided detailed enzymatic data that has shown AftD as an α -(1-5) arabinosyltransferase.

c) As a note, the authors refer to the branched termini of arabinan of AG and LAM, but they fail to acknowledge that LAM also has a linear motif Ara-4 which ends in a β -linked arabinose, through presumably AftB as an β -(1-2)-arabinosyltransferase?

This needs to be corrected in Extended Data Figure 1, and elsewhere in the manuscript, and should be discussed.

d) To confirm that recombinant AftB is active, we adapted a reported arabinosyltransferase assay 42 using a tetrasaccharide acceptor (Extended Data Fig. 2a).

It is clear that the authors have used farnesylphosphoarabinose (FPA) as an alternative substrate to DPA, based on the cited publication of Angala et al. (2021) ACS Chem. Biol. 16(1), 20, which describes the biochemistry surrounding AftA and initiation of arabinan biosynthesis? Whilst the use of FPA is fine, the actual experiments need better controls and substrates which actually define AftB as an α -arabinosyltransferase (see citation 37). For instance:

(i) Why are the authors using *E. coli* membranes expressing AftB for their assays, they also describe *M. smegmatis* membranes and assays but none were reported? They have elegantly purified the AftB protein for their structural studies, why not assay the purified AftB – this is a must as the earlier publications on Emb and Aft proteins all used purified recombinant proteins for biochemical studies, so the assays should be performed with purified AftB rather than crude *E. coli* membrane preparations expressing AftB.

(ii) Furthermore, the authors need to consider the use of appropriate controls. The assays should have at least *E. coli* membranes with an empty plasmid as a control as presented; the use of mutated AftB-Asp62 as control is not very useful, do we know the protein is correctly folded, so can't be used as a control, plus data not shown is not really helpful? This should be repeated with an empty plasmid control. The use of purified AftB in assays would simply require a +/- AftB proteins.

(iii) Without mass spectral ring-fragmentation data, the authors can't simply conclude that AftB is an β -arabinosyltransferase as presented, an increase in mass only determines that the acceptors have added one or two arabinose residues, this is not definitive evidence of β -arabinosyltransferase activity, and the structure of the products is speculative as reported with these particular acceptors and FPA.

(iv) The experiments should be repeated according to the original studies that described AftB (cited reference 37), as the simpler disaccharide substrate only serves as a substrate for two activities: α -(1-5)-arabinosyltransferase and β -(1-2)-arabinosyltransferase; the inclusion of ethambutol will inhibit the α -(1-5)-arabinosyltransferase, providing further evidence of the ethambutol resistant β -(1-2)-arabinosyltransferase activity and this will enhance the functional activity and provide further evidence of AftB as an ethambutol resistant β -(1-2)-arabinosyltransferase activity.

(v) While proposing key insights into the structure and function of AftB, no key residues suggested to be important for DPA/FPA binding and catalysis were in fact mutated and assessed further biochemically, thus validating their importance relevant to mechanistic insights. Clearly, key residues should be mutated in AftB and those proteins should be purified and assessed biochemically, this would strengthen some of their claims in terms of substrate binding and catalysis.

Minor Points

- a) Some of the text in the Figures is too small to be legible, particularly Figure 1 and extended data Figure 2.
- b) Line 63 comma after ethambutol
- c) Line 137 insert '5' after '...JM helices 4 and'
- d) Line 157 comma after AftB
- e) Line 158 delete '1 and 7' or replace with '(TM1-TM7)'
- f) Line 172 I can't see Asp62 labelled in Fig 2a.
- g) Line 175 mutant should be purified and shown to be correctly folded, perhaps with circular dichroism

Reviewer #2

(Remarks to the Author)

In this paper, Liu et al reports the structures of *Mycobacteria chubuense* AftB, a terminal arabinosyltransferase in the formation of mycobacterial cell wall arabinan containing AG and LAM. The work appears to be technically sound, in particular, it is impressive that the authors identified a conformational Fab binder for EM structure determination by using phage display and a synthetic library. The two EM structures reported in this study are of sufficient quality, what's missing, for a broad readership, is a description of how these findings will enable new antitubercular drug development. And major questions and thoughts that the authors should clarify further.

Major issue:

It is a pity that, the authors haven't shown any structure comparison with other reported mycobacterial GT-C enzymes, including the ethambutol target EmbB and EmbC.

It is encouraging that the authors determined the AftB structure from *Mycobacteria chubuense* by extensive screening work. However, given the fact that this species is very uncommonly used, careful comparison between *Mycobacteria chubuense* and *M. tuberculosis* should be provided in terms of both structural and functional analysis.

In this paper the authors used a MALDI mass spectrometry based experiment for AraT activity study. Only WT and the catalytic site mutant D62A were shown in Extended Data Figs. 2d-e. Other residues critical for substrate recognition, such as R221, R372, T91, T274 and E172 mentioned in this study have not been analyzed. It is obvious that only sequence conservation is not enough to support their roles in substrate recognition. We are not convinced of the data presented in Extended Data Figs. 2d-e, as the resolution of the spectrum in the figure is very poor, including almost all values on the X and Y axis.

In this paper the activity of AftB was detected in the presence of 10 mM MgCl₂, however they had no description on whether the activity is cation-dependent, nor had they shown an EDTA control of their AftB activity analysis. However, this is important for mechanism analysis.

It seems that the Fab used for EM study is close to the proposed AraF4 cavity and may block the substrate entrance. However, the effect of Fab was not considered during their AraT activity study. The authors should state if the Fab-B3 inhibit the AftB enzymatic activity, and clarify whether the EM structure they determined is in the inhibitory state or not. As the Fab-bound structure was used for the following MD simulation study and the authors failed to obtain a stable AraF4-/ AraF6- bound MD result.

Minor issue:

Fig. 1a, bottom: the linkage between the arabinan chain and the galactan chain is inaccurate, is it the 10th position of the galactan chain? please either label this on the figure or specify this in the legend.

Line60: the schematic representation of LAM biosynthesis is in Extended Data Fig. 1a, not together with the one of AG shown in Fig. 1a. Can the author just combine them into the same figure, such as Fig1?

Line83: please indicate AraF6 in Fig1a

Line130: contains two domains

Line137: "harbors JM helices 4 and" what? Proofread carefully...

Line165: please specify how this unique feature may "contribute to AftB's specific substrate recognition or catalytic mechanism", or tune down the words here. And the two beta strands connecting the 3rd and the 10th TMH should be appeared in the overall structure of AftB.

Line225: please label all these interacting residues in Extended Data Fig. 7.

Line277, we are not very convinced by the double-displacement mechanism of AftB proposed by the authors. As this is only supported by the position of the catalytic residue which is close to the donor's anomeric carbon. However, this is weak as they lack the important acceptor-bound structure. Can the authors just rule out any other possibility of the mechanism? If not, please describe in the discussion.

Observation of AcpM bound with mycobacterial GT-C enzymes, such as Emb proteins and AftD is an interesting discovery in the field. In this paper the structure of AftB was captured in the absence of AcpM, may the authors provide some structural insights into this finding in the discussion?

Line659: the classic disaccharide linker of the AG should be GlcNAc-Rha.

Extended Data Figs. 4 and 5, please give the contour level of each map density.

Line787, please provide map sharpening B factor.

Reviewer #3

(Remarks to the Author)

Reviewer #4

(Remarks to the Author)

In this work, Liu et al have determined AftB (an arabinofuranosyltransferase terminating the arabinan synthesis in mycobacteria) cryo-EM structures in apo and a substrate FPA bound forms, and utilized modeling and molecular dynamics (MD) simulations to better understand substrate binding.

Majors

Surprisingly, they mentioned AftA, an arabinofuranosyltransferase initiating the arabinan synthesis, in Introduction once, but did not mention its structural and functional work (<https://www.pnas.org/doi/10.1073/pnas.2302858120>) in detail in the paper, given its high similarity to AftB.

Line 174: We confirmed comparable expression levels of the D62A mutant with wild175 type McAftB (data not shown), The authors should show the data.

Fig. 3b mentioned many key residues in the active site, but there are no functional data unlike the AftA work.

The authors used the apo structure to model terminal-Araf4, Is it possible to model it in the FPA-bound structure?

A terminal-Araf6-bound model: Was the structure modeled by RoseTTAFold too? Its structure is not shown. What are the differences between Araf4 and Araf6 bound structures? Is there enough room for Araf6?

Why the authors believe that the lengthy repeating sugar units and the lipid membrane anchors will make the terminal-Araf4 binding stable? And, should DP (or DPP) be supposed to leave the site as a product?

pKa in Extended Data Fig. 10c: How this was measured or calculated? It is not mentioned in the method. Is it reasonable to have pKa ~9 of Asp62 in the apo state? Even in the substrate bound state, the pKa values are around 6 with large errors, meaning that Asp62 could be protonated a lot more than in solution.

A double-displacement mechanism in Line 277: No ions involved? Which of the authors' data actually support this mechanism? In this paragraph, there are no citations, although the authors mentioned in Discussion.

Minors

Line 184: Extended Data Fig. 9 is the simulation data and is nothing to do with binding site opening.

Line 188: Is Extended Data Fig. 8f a correct figure to mention?

Line 195: Analyses of our simulations -> Analyses of our simulations starting from randomly placed DAP in the upper leaflet

Line 198: In the CG-MD simulations -> starting from randomly placed the acceptor in bulk solvent (and remove "bulk solvent" at the end of sentence)

Line 201: For comparison, RoseTTAFold All-Atom55 was used to fold and dock both substrates and products to AftB, further predicting and characterizing their binding sites. It is strange to end the sentence like this. Should the authors mention the comparison?

Line 250: DP or DPP (used in Extended Data Fig. 8)?

Version 1:

Reviewer comments:

Reviewer #1

(Remarks to the Author)

The authors have greatly improved the manuscript through the additional data provided as requested and the manuscript is an excellent addition to the field.

No further concerns or experiments required.

Reviewer #2

(Remarks to the Author)

[Editorial Note: Please also see attachment at the end of the file for greater clarity]

The authors improved the revision by adding activity analysis of AftB mutants, and more detailed structure analysis/ comparison.

In this paper, Liu et al reports the structures of Mycobacteria chubuense AftB, a terminal arabinosyltransferase in the formation of mycobacterial cell wall arabinan containing AG and LAM. The work appears to be technically sound, in particular, it is impressive that the authors identified a conformational Fab binder for EM structure determination by using phage display and a synthetic library. The two EM structures reported in this study are of sufficient quality, what's missing, for a broad readership, is a description of how these findings will enable new antitubercular drug development. And major questions and thoughts that the authors should clarify further.

We thank the Reviewer for their positive comments. We have expanded our Discussion to include how our findings enable new antitubercular drug development. A new paragraph in the Discussion section (lines 447-461) highlights the potential of AftB as a drug target and how our structural insights could guide rational drug design.

The discussion is weak and in lack of deep structural insight, such as which pocket/ cavity is an ideal one for competitive/ noncompetitive inhibitor discovery? Otherwise the results reported in this paper is but structures. And I am confused why the authors cited Extended Data Fig. 8 in line 458, is there any information for structural comparison here?

Major issue:

It is a pity that, the authors haven't shown any structure comparison with other reported mycobacterial GT-C enzymes, including the ethambutol target EmbB and EmbC.

We thank the Reviewer for having raised this important point. In response, we have added a comprehensive structural comparison section that analyzes AftB in relation to other mycobacterial arabinosyltransferases, including AftA, AftD, and EmbB as a representative of the Emb protein family. We chose EmbB as a representative because it shares high structural similarity with EmbA and EmbC. This comparison is presented in a new supplementary figure (Supplementary Figs. 3 and 4).

Should this be added in the Extended Data Figures ?

In this paper the activity of AftB was detected in the presence of 10 mM MgCl₂, however they had no description on whether the activity is cation-dependent, nor had they shown an EDTA control of their AftB activity analysis. However, this is important for mechanism analysis.

The conditions used in the AraT assay were those used previously by other investigators and include MgCl₂; our focus was not to probe the metal ion dependence of the enzyme. A literature search to determine if the metal ion dependence of the Aft and Emb enzymes had been previously investigated revealed that, surprisingly, it appears that it has not been. We agree that this is useful information, but we see it as outside the scope of the present manuscript. Importantly, while a metal ion may be involved, we do not think it would meaningfully change our key conclusion that the enzyme proceeds through a double displacement mechanism.

This question was raised based on Fig.3b and 3d, wherein the arabinosyl-phosphate group is bound in a negatively charged cavity. So how the phosphate leaving group is activated in the transfer reaction? Metal-ion dependency is a part of the mechanism analysis.

It seems that the Fab used for EM study is close to the proposed Araf4 cavity and may block the substrate entrance.

However, the effect of Fab was not considered during their AraT activity study. The authors should state if the Fab-B3 inhibits the AftB enzymatic activity, and clarify whether the EM structure they determined is in the inhibitory state or not. As the Fab-bound structure was used for the following MD simulation study and the authors failed to obtain a stable Araf4-/ Araf6- bound MD result.

While our atomistic simulations with terminal-Araf4/6 are unstable, the CG simulations with the apo state without the Fab show clear binding. When we performed CG simulations with Araf4 and the Fab present, there was still binding of the sugar to the proposed binding site, consistent with the fact that the Fab does not inhibit acceptor binding (Extended Data Fig.10a (ii)).

Additionally, RMSF analysis revealed that protein dynamics remain similar across apo, Fab-bound, substrate-bound, and product-bound states (Extended Data Fig.10a (i)), indicating that Fab does not significantly alter protein flexibility. We have added these analyses to the Results section (lines 236-240).

Fab does not inhibit acceptor binding is supported by *in silico* analysis, rather than “the fact”. From the EM data, Fab doesn’t significantly interfere with the proposed acceptor entrance.

An easy way is to measure the AftB enzymatic activity in the presence of Fab, it will be perfect to introduce Fab in the activity assay established by the authors.

Line277, we are not very convinced by the double-displacement mechanism of AftB proposed by the authors. As this is only supported by the position of the catalytic residue which is close to the donor’s anomeric carbon. However, this is weak as they lack the important acceptor-bound structure. Can the authors just rule out any other possibility of the mechanism? If not, please describe in the discussion.

We too were disappointed that we could not obtain any complexes with an acceptor bound and that the fluorinated FPA did not produce a stable enzyme-linked intermediate. Because this is the first example of a retaining GT-C enzyme to be structurally-characterized, drawing parallels with other enzymes is impossible. To address this point, we enhanced the discussion (lines 384–412) better comparing similarities to the structure of WbbB (a GT-B enzyme), which is one of two known retaining glycosyltransferases demonstrated to proceed via a double displacement mechanism and also retaining glycosidases, which proceed by a similar mechanism. It should be appreciated that the key element of data leading to the suggestion (and now acceptance) of the alternate S_Ni mechanism for most retaining (GT-A and GT-B) glycosyltransferases, is the lack of a catalytic residue appropriately placed to act as a nucleophile. Such a residue is present in AftB, and the distance to the anomeric center in the donor is identical to a donor-bound complex of WbbB and in the same range of that typically found in retaining glycosidases.

Still not fully convinced. Suggestion: Combine and move the “catalytic mechanism of AftB” to the “mechanistic insights into glycosyl transfer” in the discussion.

Other:

Extended Fig. 6 e, the panel was AftA, while the caption was STT3B.

Supplementary Figure 4 c, do you mean Mtb AftA from PDB databank or Ms AftA from a predicted model? There is no reported structure of Ms AftA that I can find.

Reviewer #3

(Remarks to the Author)

Reviewer #4

(Remarks to the Author)

The authors have addressed all my comments and questions properly.

Response to Reviewers

Liu et al. “Structural insights into terminal arabinosylation biosynthesis of the mycobacterial cell wall arabinan”

We are thankful for the overall positive and constructive feedback from the Reviewers. Here, we provide a point-by-point response to their comments.

Reviewer #1:

The manuscript by Liu et al. describes insights into the terminal arabinosylation biosynthesis of the mycobacterial cell wall. It follows on from earlier publications which have characterized the detailed structural biology (by Cryo-EM and X-ray crystallography) and biochemistry of the related Mycobacterium tuberculosis arabinosyltransferases EmbA, B and C proteins reported by Zhang et al. in Science (2020) 368(6496), 1211, and Protein Science (2020) 11(7), 505, AftD by Tan et al. in Molecular Cell (2020) 78(4), 683, and AftA by Gong et al. in PNAS (2023) 120(23), e2302858120.

At first glance, it is unclear how the current manuscript on AftB, other than being a retaining β -arabinosyltransferase, whereas the others are inverting α -arabinosyltransferases, based on the donor decaprenolphosphoarabinose (DPA), provides a conceptual advance to this class of GT-C enzymes, and in particular due to the lack of biochemical data linked to the analysis of purified mutated AftB proteins, which would unequivocally demonstrate activity and mechanistic insights into AftB.

Major Points

It was notable that two key errors describing areas relevant to the present literature have been made in the current manuscript relevant to α -arabinosyltransferases, which must be corrected. In addition, the biochemical data is insufficient evidence in confirming that AftB is an β -arabinosyltransferase, although it is known to be in the original cited literature. Furthermore, the authors should extend their analyses to include mutated purified AftB proteins to probe the mechanism and function of AftB.

We acknowledge these points and have addressed them as follows:

1. We have corrected the literature errors regarding mycobacterial α -arabinosyltransferases.
2. We have obtained additional biochemical data, including MS/MS data of reaction products, to confirm the regioselectivity of AftB.
3. We have conducted mutagenesis studies on several key residues and performed biochemical assays with *E. coli* membrane expressing these mutant proteins R211A, R211E, R372A, R372E, F448A and Y449A in addition to D62A (in Supplementary Fig. 2). While the assays for these mutants were successful, we acknowledge that MS analysis of the products presented challenges due to weak signal intensity. The MS method we have used is one of two reported assays for mycobacterial AraTs; the other employs radiolabeled DPA donor substrate and quantitation using scintillation counting of labeled products. The latter is inherently more sensitive, but is complicated by the need to employ radiochemical techniques, in both the assay and in the preparation of the DPA substrate. We have therefore opted for the less sensitive, but more experimentally feasible MS approach.

Nevertheless, the data confirms their critical roles in substrate recognition and catalysis, which is also supported by computational modelling. These results are presented in Supplementary Fig. 2a and described in the Result sections (lines 260-262 and 309-311).

a) Line 74 – This is followed by the action of enzymes EmbA and EmbB, which extend the arabinan chain through α -(1-5) linkages 31 (Fig. 1a).

This is factually incorrect. If the authors were to re-read citation (31, Escuyer et al. [2001] J. Biol. Chem. 276(52), 48854) as listed in the references list, it was suggestive that EmbA and EmbB were hypothesised as α -(1-3) linkages involved in the terminal stages of arabinan synthesis and NOT α -(1-5) linkages. The definitive evidence really came from the recent Zhang et al. publication in Science (2020) 368(6496), 1211, which clearly demonstrated that EmbA/B formed a novel protein complex and the activity was indeed an α -(1-3) linkage at the terminal stages of arabinan synthesis. This sentence needs to be corrected based on findings in the citation of Zhang et al. as the present citation is outdated as it refers to literature which was speculative at the time, and the correct citation of Zhang et al. used at the appropriate places in the manuscript.

We apologize for this error. We have corrected the manuscript to accurately describe the function of EmbA and EmbB, citing Zhang et al. (2020) to reflect that they form α -(1 \rightarrow 3) linkages at the terminal stages of arabinan synthesis (lines 74-75).

b) AftD is thought to have a similar function to AftC, yet its precise role remains not fully elucidated 34, 35. This is again factually incorrect. The authors need to refer to the citation of Alderwick et al. (2018) Cell Surf. 1, 2, which provided detailed enzymatic data that has shown AftD as an α -(1-5) arabinosyltransferase.

We have corrected the statement about AftD's role, citing Alderwick et al. (2018) to accurately describe AftD as an α -(1 \rightarrow 5) arabinosyltransferase (lines 78-80).

c) As a note, the authors refer to the branched termini of arabinan of AG and LAM, but they fail to acknowledge that LAM also has a linear motif Ara-4 which ends in a β -linked arabinose, through presumably AftB as an β (1-2)-arabinosyltransferase? This needs to be corrected in Extended Data Figure 1, and elsewhere in the manuscript, and should be discussed.

We thank the Reviewer for highlighting this aspect of LAM structure. We did touch upon this in our original manuscript as part of the Introduction and Discussion (lines 59-60 and 305-309). However, we recognize that this point may not have been emphasized sufficiently. Our study primarily focuses on describing the branched acceptor substrate motifs, which represent a key shared feature of AG and LAM and were central to our computational analyses. However, we agree that the linear Ara₄ motif merits discussion. We have added text in the Introduction, Results and Discussion sections to explicitly address LAM's linear Ara₄ motifs (lines 84-86, 219-223, and 375-380). Additionally, we have mentioned in the Discussion section that future studies are necessary to provide further insight into the structural basis of AftB's recognition of both branched and linear arabinan substrates (lines 381-382). We have also updated Extended Data Fig. 1 to highlight the linear Ara₄ motif in LAM that ends with a β -linked arabinose residue.

d) To confirm that recombinant AftB is active, we adapted a reported arabinosyltransferase assay 42 using a tetrasaccharide acceptor (Extended Data Fig. 2a).

It is clear that the authors have used farnesylphosphoarabinose (FPA) as an alternative substrate to DPA, based on the cited publication of Angala et al. (2021) ACS Chem. Biol. 16(1), 20, which

describes the biochemistry surrounding AftA and initiation of arabinan biosynthesis? Whilst the use of FPA is fine, the actual experiments need better controls and substrates which actually define AftB as an β -arabinosyltransferase (see citation 37). For instance:

(i) Why are the authors using E. coli membranes expressing AftB for their assays, they also describe M. smegmatis membranes and assays but none were reported? They have elegantly purified the AftB protein for their structural studies, why not assay the purified AftB – this is a must as the earlier publications on Emb and Aft proteins all used purified recombinant proteins for biochemical studies, so the assays should be performed with purified AftB rather than crude E. coli membrane preparations expressing AftB.

While we acknowledge that previous structural studies on Emb and several other Aft proteins have used purified recombinant proteins for biochemical assays, the original characterization of AftB (PMID: 17387176) employed membrane fractions to assess its activity. We used *E. coli* membranes expressing AftB and *M. smegmatis* membranes to demonstrate AftB activity in different contexts. The *M. smegmatis* membranes served as a control, while the *E. coli* membranes expressing wild-type AftB proved that the AftB we studied is active. When we attempted to assay the purified AftB protein, we could not detect activity outside of the membrane environment, which may suggest that the membrane may be required for AftB's activity.

(ii) Furthermore, the authors need to consider the use of appropriate controls. The assays should have at least E. coli membranes with an empty plasmid as a control as presented; the use of mutated AftB-Asp62 as control is not very useful, do we know the protein is correctly folded, so can't be used as a control, plus data not shown is not really helpful? This should be repeated with an empty plasmid control. The use of purified AftB in assays would simply require a +/- AftB proteins.

We have ensured that the D62A mutant is properly folded by assessing its expression levels and mono-dispersity (standard way to relate to proper folding) using Size Exclusion Chromatography (SEC). These data are now included in Supplementary Fig. 1. We did not do an empty plasmid control. While we acknowledge that an empty plasmid control would provide additional validation, the complete loss of activity in assays with all the mutants, combined with their proper folding and expression, strongly suggests that the observed activity in wild-type samples is specifically due to AftB function rather than background *E. coli* activities.

(iii) Without mass spectral ring-fragmentation data, the authors can't simply conclude that AftB is an β -arabinosyltransferase as presented, an increase in mass only determines that the acceptors have added one or two arabinose residues, this is not definitive evidence of β -arabinosyltransferase activity, and the structure of the products is speculative as reported with these particular acceptors and FPA.

We have provided MS/MS data of the products using the tetrasaccharide acceptor confirming the (1 \rightarrow 2)-linkage in both the pentasaccharides and the hexasaccharide (Supplementary Fig. 2b-c), and provided a detailed description of our analysis in the Result section (lines 108-148). Insufficient materials were available to carry out ^1H NMR analysis of these compounds to unequivocally confirm the β -stereochemistry. However, because no α -Araf-(1 \rightarrow 2)-Araf linkages have been reported in mycobacterial arabinans, we are confident in our stereochemical assignment. We note that the majority of previous papers functionally characterizing mycobacterial AraTs using synthetic acceptor substrates have correlated MS (and/or methylation analysis) data with

what is known about the structure of mycobacterial arabinan to demonstrate linkage and stereochemistry.

(iv) The experiments should be repeated according to the original studies that described AftB (cited reference 37), as the simpler disaccharide substrate only serves as a substrate for two activities: α -(1-5)-arabinosyltransferase and β -(1-2)-arabinosyltransferase; the inclusion of ethambutol will inhibit the α -(1-5)-arabinosyltransferase, providing further evidence of the ethambutol resistant β -(1-2)-arabinosyltransferase activity and this will enhance the functional activity and provide further evidence of AftB as an ethambutol resistant β -(1-2)-arabinosyltransferase activity.

The MS/MS data obtained on the products using the tetrasaccharide, in our opinion make the requested experiment redundant. We further explain (lines 100–102 and 108–112) why we chose to use the tetrasaccharide acceptor. We appreciate the Reviewer's suggestion, but we consider re-confirming the ethambutol resistance of AftB to be beyond the scope of this manuscript.

(v) While proposing key insights into the structure and function of AftB, no key residues suggested to be important for DPA/FPA binding and catalysis were in fact mutated and assessed further biochemically, thus validating their importance relevant to mechanistic insights. Clearly, key residues should be mutated in AftB and those proteins should be purified and assessed biochemically, this would strengthen some of their claims in terms of substrate binding and catalysis.

Data for additional mutants are provided as detailed above. These data show that the mutants prepared are inactive in the assay we have used, consistent with the interactions we observed in our structures, and further supported by our computational work.

Minor Points

a) Some of the text in the Figures is too small to be legible, particularly Figure 1 and extended data Figure 2.

We have increased the font size in Fig. 1 and Extended Data Fig. 2 to improve legibility.

b) Line 63 comma after ethambutol

This correction has been made (line 62).

c) Line 137 insert '5' after '...JM helices 4 and'

This correction has been made. (now in line 177)

d) Line 157 comma after AftB

We have corrected this and edited the original sentence (line 197).

e) Line 158 delete '1 and 7' or replace with '(TM1-TM7)'

We have replaced '1 and 7' with '1 to 7' for clarity (line 197).

f) Line 172 I can't see Asp62 labelled in Fig 2a.

We have labeled Asp62 in Fig. 2a.

g) Line 175 mutant should be purified and shown to be correctly folded, perhaps with circular dichroism

We have purified the D62A mutant and confirmed it is folded. This data has been included in Supplementary Fig. 1.

Reviewer #2:

In this paper, Liu et al reports the structures of Mycobacteria chubuense AftB, a terminal arabinosyltransferase in the formation of mycobacterial cell wall arabinan containing AG and LAM. The work appears to be technically sound, in particular, it is impressive that the authors identified a conformational Fab binder for EM structure determination by using phage display and a synthetic library. The two EM structures reported in this study are of sufficient quality, what's missing, for a broad readership, is a description of how these findings will enable new antitubercular drug development. And major questions and thoughts that the authors should clarify further.

We thank the Reviewer for their positive comments. We have expanded our Discussion to include how our findings enable new antitubercular drug development. A new paragraph in the Discussion section (lines 447-461) highlights the potential of AftB as a drug target and how our structural insights could guide rational drug design.

Major issue:

It is a pity that, the authors haven't shown any structure comparison with other reported mycobacterial GT-C enzymes, including the ethambutol target EmbB and EmbC.

We thank the Reviewer for having raised this important point. In response, we have added a comprehensive structural comparison section that analyzes AftB in relation to other mycobacterial arabinosyltransferases, including AftA, AftD, and EmbB as a representative of the Emb protein family. We chose EmbB as a representative because it shares high structural similarity with EmbA and EmbC. This comparison is presented in a new supplementary figure (Supplementary Figs. 3 and 4).

It is encouraging that the authors determined the AftB structure from Mycobacteria chubuense by extensive screening work. However, given the fact that this species is very uncommonly used, careful comparison between Mycobacteria chubuense and M.tuberculosis should be provided in terms of both structural and functional analysis.

We thank the Reviewer for raising this important point. We have added a new section in the Results titled "Structural conservation across mycobacterial species" (lines 265-277). We compared our structure with AlphaFold2 predictions of AftB from *M. tuberculosis*, the causative agent of tuberculosis, and *M. smegmatis*, a widely established model organism used in mycobacterial biology. This comparison, illustrated in Supplementary Fig. 2, demonstrates high structural conservation, particularly in the substrate binding cavity and catalytic domains. From a functional perspective, we demonstrated that our recombinant *M. chubuense* AftB is enzymatically active using the established arabinosyltransferase assay system, catalyzing the same β -(1 \rightarrow 2) arabinosyl transfer reaction as reported for mycobacterial AftB. The conservation of catalytic residues, combined with our functional validation, strongly suggests that the catalytic mechanism we have elucidated for *M. chubuense* AftB reflects that of *M. tuberculosis* AftB as well.

In this paper the authors used a MALDI mass spectrometry based experiment for AraT activity study. Only WT and the catalytic site mutant D62A were shown in Extended Data Figs. 2d-e. Other

residues critical for substrate recognition, such as R221, R372, T91, T274 and E172 mentioned in this study have not been analyzed. It is obvious that only sequence conservation is not enough to support their roles in substrate recognition. We are not convinced of the data presented in Extended Data Figs. 2d-e, as the resolution of the spectrum in the figure is very poor, including almost all values on the X and Y axis.

As we described in the response to Reviewer 1, we have conducted mutagenesis studies on several key residues and performed biochemical assays with *E. coli* membrane expressing these mutant proteins R211A, R211E, R372A, R372E, F448A and Y449A in addition to D62A. The MALDI mass spectrometry data for D62A and all the additional mutants are included in Supplementary Fig. 2. We have improved the resolution of the spectra.

In this paper the activity of AftB was detected in the presence of 10 mM MgCl₂, however they had no description on whether the activity is cation-dependent, nor had they shown an EDTA control of their AftB activity analysis. However, this is important for mechanism analysis.

The conditions used in the AraT assay were those used previously by other investigators and include MgCl₂; our focus was not to probe the metal ion dependence of the enzyme. A literature search to determine if the metal ion dependence of the Aft and Emb enzymes had been previously investigated revealed that, surprisingly, it appears that it has not been. We agree that this is useful information, but we see it as outside the scope of the present manuscript. Importantly, while a metal ion may be involved, we do not think it would meaningfully change our key conclusion that the enzyme proceeds through a double displacement mechanism.

It seems that the Fab used for EM study is close to the proposed Araf₄ cavity and may block the substrate entrance. However, the effect of Fab was not considered during their AraT activity study. The authors should state if the Fab-B3 inhibits the AftB enzymatic activity, and clarify whether the EM structure they determined is in the inhibitory state or not. As the Fab-bound structure was used for the following MD simulation study and the authors failed to obtain a stable Araf₄-/Araf₆-bound MD result.

While our atomistic simulations with terminal-Araf_{4/6} are unstable, the CG simulations with the apo state without the Fab show clear binding. When we performed CG simulations with Araf₄ and the Fab present, there was still binding of the sugar to the proposed binding site, consistent with the fact that the Fab does not inhibit acceptor binding (Extended Data Fig. 10a (ii)).

Additionally, RMSF analysis revealed that protein dynamics remain similar across apo, Fab-bound, substrate-bound, and product-bound states (Extended Data Fig. 10a (i)), indicating that Fab does not significantly alter protein flexibility. We have added these analyses to the Results section (lines 236-240).

Minor issue:

Fig. 1a, bottom: the linkage between the arabinan chain and the galactan chain is inaccurate, is it the 10th position of the galactan chain? please either label this on the figure or specify this in the legend.

While we acknowledge that this linkage can also occur at the 8th or 12th position, we have depicted the 10th position for clarity and simplicity. We have updated the figure legend to specify this detail explicitly.

Line60: the schematic representation of LAM biosynthesis is in Extended Data Fig. 1a, not together with the one of AG shown in Fig. 1a. Can the author just combine them into the same figure, such as Fig1?

We appreciate the suggestion to combine the schematic representations of AG and LAM biosynthesis into Fig. 1. While we understand that this could provide a more comprehensive view, integrating both pathways into a single figure would make it overly crowded and reduce clarity. We chose to separate the two figures to focus on the specific role of AftB within AG biosynthesis in Fig. 1a, while presenting LAM biosynthetic pathway in Extended Data Fig. 1a. We hope this approach ensures both figures remain clear and easy to interpret.

Line83: please indicate Araf6 in Fig1a

We have labeled Araf6 in Fig. 1a.

Line130: contains two domains

This correction has been made (line 169).

Line137: "harbors JM helices 4 and" what? Proofread carefully...

We have corrected this sentence (line 177).

Line165: please specify how this unique feature may "contribute to AftB's specific substrate recognition or catalytic mechanism", or tune down the words here. And the two beta strands connecting the 3rd and the 10th TMH should be appeared in the overall structure of AftB.

We have moderated our description of this structural feature's potential functional implications (line 202-203).

Line225: please label all these interacting residues in Extended Data Fig. 7.

We have labeled all interacting residues.

Line277, we are not very convinced by the double-displacement mechanism of AftB proposed by the authors. As this is only supported by the position of the catalytic residue which is close to the donor's anomeric carbon. However, this is weak as they lack the important acceptor-bound structure. Can the authors just rule out any other possibility of the mechanism? If not, please describe in the discussion.

We too were disappointed that we could not obtain any complexes with an acceptor bound and that the fluorinated FPA did not produce a stable enzyme-linked intermediate. Because this is the first example of a retaining GT-C enzyme to be structurally-characterized, drawing parallels with other enzymes is impossible. To address this point, we enhanced the discussion (lines 384–412) better comparing similarities to the structure of WbbB (a GT-B enzyme), which is one of two known retaining glycosyltransferases demonstrated to proceed via a double displacement mechanism and also retaining glycosidases, which proceed by a similar mechanism. It should be appreciated that the key element of data leading to the suggestion (and now acceptance) of the alternate S_Ni mechanism for most retaining (GT-A and GT-B) glycosyltransferases, is the lack of a catalytic residue appropriately placed to act as a nucleophile. Such a residue is present in AftB, and the distance to the anomeric center in the donor is identical to a donor-bound complex of WbbB and in the same range of that typically found in retaining glycosidases.

Observation of AcpM bound with mycobacterial GT-C enzymes, such as Emb proteins and AftD is an interesting discovery in the field. In this paper the structure of AftB was captured in the absence of AcpM, may the authors provide some structural insights into this finding in the discussion?

We thank the Reviewer for this insightful observation. Our Discussion section has been expanded to address the differential ACP interactions among mycobacterial arabinosyltransferases (lines 436-445). The absence or presence of ACP binding features outside the conserved GT-C motifs likely reflects an evolutionary adaptation to different regulatory requirements. However, direct experimental validation of this hypothesis is needed. We have addressed this point through structural comparison in the Discussion.

Line659: the classic disaccharide linker of the AG should be GlcNAc-Rha.

We corrected this.

Extended Data Figs. 4 and 5, please give the contour level of each map density.

We have added this information in the figure legends of Extended Figs. 4 and 5.

Line787, please provide map sharpening B factor.

The sharpening B factor was added to Table

Reviewer #3:

Reviewer #4:

In this work, Liu et al have determined AftB (an arabinofuranosyltransferase terminating the arabinan synthesis in mycobacteria) cryo-EM structures in apo and a substrate FPA bound forms, and utilized modeling and molecular dynamics (MD) simulations to better understand substrate binding.

Majors

Surprisingly, they mentioned AftA, an arabinofuranosyltransferase initiating the arabinan synthesis, in Introduction once, but did not mention its structural and functional work (<https://www.pnas.org/doi/10.1073/pnas.2302858120>) in detail in the paper, given its high similarity to AftB.

We have now expanded our Discussion to include a comprehensive structural comparison between AftB and other mycobacterial arabinosyltransferases, specifically focusing on AftA, AftD, and EmbB as a representative of the Emb protein family (lines 413-445). This comparison examines both the structural similarities and distinct architectural features that reflect their specialized functions in arabinan biosynthesis.

Line 174: We confirmed comparable expression levels of the D62A mutant with wild175 type McAftB (data not shown), The authors should show the data.

We have now added a supplementary figure showing the expression levels of wild-type and D62A mutant McAftB (Supplementary Fig. 1).

Fig. 3b mentioned many key residues in the active site, but there are no functional data unlike the AftA work.

As described in the response to Reviewers 1 and 2, we have provided additional mutant data. Specifically, we have conducted mutagenesis studies on several key residues and performed biochemical assays with *E. coli* membrane expressing these mutant proteins R211A, R211E, R372A, R372E, F448A and Y449A in addition to D62A (in Supplementary Fig. 2). To verify proper protein folding and expression levels, we conducted size exclusion chromatography and SDS-PAGE analyses of all mutants. These results are presented in Supplementary Figs. 1 and 2a and described in the Result sections (lines 260-262 and 309-311).

The authors used the apo structure to model terminal-Araf4, Is it possible to model it in the FPA-bound structure?

The RoseTTAFold All-atom approach, shown in Extended Data Fig. 8, and our subsequent Chai-1 modelling, does not take into account the structure of AftB, rather modelling based on the amino acid sequence and provided SMILES string. Nevertheless, it is also possible to use ligand docking to propose binding orientations for the substrates.

A terminal-Araf6-bound model: Was the structure modeled by RoseTTAFold too? Its structure is not shown. What are the differences between Araf4 and Araf6 bound structures? Is there enough room for Araf6?

We have modelled terminal-Araf6 with both RoseTTAFold and Chai-1. There are limited differences in AftB between the two. The primary difference is that the terminal Araf is always modelled at the same position for both substrate and product, suggesting that the terminal-Araf6 is modelled in a position ready to leave the active site.

Why the authors believe that the lengthy repeating sugar units and the lipid membrane anchors will make the terminal-Araf4 binding stable? And, should DP (or DPP) be supposed to leave the site as a product?

The lipid moiety likely acts as an anchoring point within the membrane, ensuring that the polysaccharide chain is properly oriented toward the enzyme. This anchoring helps increase the local concentration of the substrate near the active site, facilitating more efficient interactions. The longer sugar chain could also increase interactions with the protein outside of the binding site, which could act to stabilize the terminus located in the binding site. DP should be able to leave the binding site after transfer of arabinose.

pKa in Extended Data Fig. 10c: How this was measured or calculated? It is not mentioned in the method. Is it reasonable to have pKa ~9 of Asp62 in the apo state? Even in the substrate bound state, the pKa values are around 6 with large errors, meaning that Asp62 could be protonated a lot more than in solution

The calculation method has been incorporated into the catalytic mechanism results section (lines 338-343) and the MD methods (lines 720-722). Using the propkatraj tool, we measured the pKa of residues at every time point in the simulation, resulting in a wide spread of data points. A pKa value of 9 suggests that this residue would predominantly exist in a protonated state under these conditions; however, the pKa decreases with the presence of substrates or products. While these values indicate that a fraction of the population will remain protonated, a significant proportion will be deprotonated, rendering it available for its role in the transfer mechanism.

A double-displacement mechanism in Line 277: No ions involved? Which of the authors' data actually support this mechanism? In this paragraph, there are no citations, although the authors mentioned in Discussion.

This point has been addressed above in the responses to Reviewer 2, but to summarize, we have shown that the key catalytic residue, D62, is appropriately placed in the donor-bound structure and that mutating this residue to alanine abrogates activities. We have no evidence that a metal is required for catalysis, however the proximity of conserved residues R221, K278 and R372 to the active site make divalent cation binding less likely. In addition, neither RoseTTAFold All-Atom or Chai-1 proposed a binding site for metal ions.

Minors

Line 184: Extended Data Fig. 9 is the simulation data and is nothing to do with binding site opening.

We have corrected the reference to the appropriate figure that shows the binding site opening.

Line 188: Is Extended Data Fig. 8f a correct figure to mention?

Thank you for catching this. We have double-checked and corrected the figure reference (lines 219-220).

Line 195: Analyses of our simulations -> Analyses of our simulations starting from randomly placed DAP in the upper leaflet

We have made this correction as suggested (lines 229-231).

Line 198: In the CG-MD simulations -> starting from randomly placed the acceptor in bulk solvent (and remove "bulk solvent" at the end of sentence)

We have made this change as suggested (lines 232-234).

Line 201: For comparison, RoseTTAFold All-Atom55 was used to fold and dock both substrates and products to AftB, further predicting and characterizing their binding sites. It is strange to end the sentence like this. Should the authors mention the comparison?

We have updated this section (lines 234-240) of the manuscript and thank the reviewers for highlighting.

Line 250: DP or DPP (used in Extended Data Fig. 8)?

We have standardized the terminology throughout the manuscript to use DP (decaprenyl phosphate) for clarity.

Response to Reviewers

Liu et al. “Structural insights into terminal arabinosylation biosynthesis of the mycobacterial cell wall arabinan”

Reviewer #1:

The authors have greatly improved the manuscript through the additional data provided as requested and the manuscript is an excellent addition to the field.

Reviewer #3:

Reviewer #4:

The authors have addressed all my comments and questions properly. No further concerns or experiments required.

We are grateful to all the Reviewers for their appreciation of our work and thank them again for their constructive criticisms that have been truly important for us to be able to improve the manuscript.

Reviewer #2:

The authors improved the revision by adding activity analysis of AftB mutants, and more detailed structure analysis/ comparison.

In this paper, Liu et al reports the structures of Mycobacteria chubuense AftB, a terminal arabinosyltransferase in the formation of mycobacterial cell wall arabinan containing AG and LAM. The work appears to be technically sound, in particular, it is impressive that the authors identified a conformational Fab binder for EM structure determination by using phage display and a synthetic library. The two EM structures reported in this study are of sufficient quality, what's missing, for a broad readership, is a description of how these findings will enable new antitubercular drug development. And major questions and thoughts that the authors should clarify further.

We thank the Reviewer for their positive comments. We have expanded our Discussion to include how our findings enable new antitubercular drug development. A new paragraph in the Discussion section (lines 447-461) highlights the potential of AftB as a drug target and how our structural insights could guide rational drug design.

The discussion is weak and in lack of deep structural insight, such as which pocket/ cavity is an ideal one for competitive/ noncompetitive inhibitor discovery? Otherwise the results reported in this paper is but structures. And I am confused why the authors cited Extended Data Fig. 8 in line 458, is there any information for structural comparison here?

We thank the Reviewer for the feedback on the discussion. Regarding the need for deeper structural insights about druggable pockets, we have made adjustments to the Therapeutic Potential section accordingly. Our apologies for the error about the citation to Extended Data Fig. 8, we have corrected this reference to Supplementary Fig. 8 and Supplementary Fig. 14.

We have discussed the implications of our findings as thoroughly as possible within the scope of our current results. We acknowledge the limitations of our work, particularly regarding detailed druggability assessments, which would be valuable areas for future investigation. Such discussions extend beyond the evidence provided by our current data, and we do not wish to engage in speculation. Our discussion is therefore focused on conclusions supported by our structural, functional, and computational analyses while acknowledging areas where future studies would be needed.

Major issue:

It is a pity that, the authors haven't shown any structure comparison with other reported mycobacterial GT-C enzymes, including the ethambutol target EmbB and EmbC.

We thank the Reviewer for having raised this important point. In response, we have added a comprehensive structural comparison section that analyzes AftB in relation to other mycobacterial arabinosyltransferases, including AftA, AftD, and EmbB as a representative of the Emb protein family. We chose EmbB as a representative because it shares high structural similarity with EmbA and EmbC. This comparison is presented in a new supplementary figure (Supplementary Figs. 3 and 4).

Should this be added in the Extended Data Figures ?

We have combined all Extended Data Figures and Supplementary Figures into the Supplementary Figures section in accordance with the journal's formatting requirements. The structural comparison with other mycobacterial GT-C enzymes has now been included as Supplementary Figure 15.

In this paper the activity of AftB was detected in the presence of 10 mM MgCl₂, however they had no description on whether the activity is cation-dependent, nor had they shown an EDTA control of their AftB activity analysis. However, this is important for mechanism analysis.

The conditions used in the AraT assay were those used previously by other investigators and include MgCl₂; our focus was not to probe the metal ion dependence of the enzyme. A literature search to determine if the metal ion dependence of the Aft and Emb enzymes had been previously investigated revealed that, surprisingly, it appears that it has not been. We agree that this is useful information, but we see it as outside the scope of the present manuscript. Importantly, while a metal ion may be involved, we do not think it would meaningfully change our key conclusion that the enzyme proceeds through a double displacement mechanism.

This question was raised based on Fig.3b and 3d, wherein the arabinosyl-phosphate group is bound in a negatively charged cavity. So how the phosphate leaving group is activated in the transfer reaction? Metal-ion dependency is a part of the mechanism analysis.

We thank the Reviewer for this question. Our structure reveals that the phosphate group is coordinated by two conserved positively charged arginine residues (R221 and R372). These arginines may provide electrostatic stabilization of the phosphate of the donor substrate, facilitating its role as a leaving group during catalysis. Mutational studies confirm the essential nature of these residues, as R221A, R221E, R372A, and R372E variants were all catalytically inactive in our functional assays. Furthermore, our molecular dynamics analyses provided further insight into functions of R221 and R372. We have updated our section for the “catalytic mechanism of AftB” in our Result section. It has been reported that some GT-C glycosyltransferases, such as ALG6 and AftA, do not require metal ions for their catalytic activity. In the discussion section of "Mechanistic Insights into Glycosyl Transfer" we have added that future studies would be valuable to further explore potential metal-ion contributions to catalysis.

It seems that the Fab used for EM study is close to the proposed AraF4 cavity and may block the substrate entrance. However, the effect of Fab was not considered during their AraT activity study. The authors should state if the Fab-B3 inhibit the AftB enzymatic activity, and clarify whether the EM structure they determined is in the inhibitory state or not. As the Fab-bound structure was used for the following MD simulation study and the authors failed to obtain a stable AraF4-/AraF6-bound MD result.

While our atomistic simulations with terminal-Araf4/6 are unstable, the CG simulations with the apo state without the Fab show clear binding. When we performed CG simulations with AraF4 and the Fab present, there was still binding of the sugar to the proposed binding site, consistent with the fact that the Fab does not inhibit acceptor binding (Extended Data Fig.10a (ii)).

Additionally, RMSF analysis revealed that protein dynamics remain similar across apo, Fab-bound, substrate-bound, and product-bound states (Extended Data Fig.10a (i)), indicating that Fab does not significantly alter protein flexibility. We have added these analyses to the Results section (lines 236-240).

Fab does not inhibit acceptor binding is supported by in silico analysis, rather than “the fact”. From the EM data, Fab doesn’t significantly interfere with the proposed acceptor entrance.

An easy way is to measure the AftB enzymatic activity in the presence of Fab, it will be perfect to introduce Fab in the activity assay established by the authors.

Our structural analysis shows that Fab-B3 binds to the periplasmic domain at a site that is distinct from the predicted acceptor substrate binding site on AftB, with no obvious steric hindrance between the two. We appreciate the suggestion to include Fab in our activity assay. While this would indeed provide direct experimental validation, it would introduce additional complexity to our membrane fraction-based assay system. In addition, the Fab has no physiological relevance to the activity of the protein; it is used only to facilitate the cryo-EM work, an approach used often in structural biology. Had our simulations suggested structural differences between the Fab-bound

and Fab-free states, such an experiment could have been warranted. We are confident that the combination of our structural observations and computational analyses provides strong evidence that Fab binding does not significantly interfere with the structure of the protein and thus represents the form present in nature.

Line277, we are not very convinced by the double-displacement mechanism of AftB proposed by the authors. As this is only supported by the position of the catalytic residue which is close to the donor's anomeric carbon. However, this is weak as they lack the important acceptor-bound structure. Can the authors just rule out any other possibility of the mechanism? If not, please describe in the discussion.

We too were disappointed that we could not obtain any complexes with an acceptor bound and that the fluorinated FPA did not produce a stable enzyme-linked intermediate. Because this is the first example of a retaining GT-C enzyme to be structurally-characterized, drawing parallels with other enzymes is impossible. To address this point, we enhanced the discussion (lines 384–412) better comparing similarities to the structure of WbbB (a GT-B enzyme), which is one of two known retaining glycosyltransferases demonstrated to proceed via a double displacement mechanism and also retaining glycosidases, which proceed by a similar mechanism. It should be appreciated that the key element of data leading to the suggestion (and now acceptance) of the alternate S_Ni mechanism for most retaining (GT-A and GT-B) glycosyltransferases, is the lack of a catalytic residue appropriately placed to act as a nucleophile. Such a residue is present in AftB, and the distance to the anomeric center in the donor is identical to a donor-bound complex of WbbB and in the same range of that typically found in retaining glycosidases.

Still not fully convinced. Suggestion: Combine and move the “catalytic mechanism of AftB” to the “mechanistic insights into glycosyl transfer” in the discussion.

We thank the Reviewer for this suggestion. We have now combined the 'catalytic mechanism of AftB' section with the 'mechanistic insights into glycosyl transfer' section in the discussion as recommended.

Other:

Extended Fig. 6 e, the panel was AftA, while the caption was STT3B.

Supplementary Figure 4 c, do you mean Mtb AftA from PDB databank or Ms AftA from a predicted model? There is no reported structure of Ms AftA that I can find.

Thanks for pointing it out. We have corrected the content.

The authors improved the revision by adding activity analysis of AftB mutants, and more detailed structure analysis/ comparison.

In this paper, Liu et al reports the structures of Mycobacteria chubuense AftB, a terminal arabinosyltransferase in the formation of mycobacterial cell wall arabinan containing AG and LAM. The work appears to be technically sound, in particular, it is impressive that the authors identified a conformational Fab binder for EM structure determination by using phage display and a synthetic library. The two EM structures reported in this study are of sufficient quality, what's missing, for a broad readership, is a description of how these findings will enable new antitubercular drug development. And major questions and thoughts that the authors should clarify further.

We thank the Reviewer for their positive comments. We have expanded our Discussion to include how our findings enable new antitubercular drug development. A new paragraph in the Discussion section (lines 447-461) highlights the potential of AftB as a drug target and how our structural insights could guide rational drug design.

The discussion is weak and in lack of deep structural insight, such as which pocket/ cavity is an ideal one for competitive/ noncompetitive inhibitor discovery? Otherwise the results reported in this paper is but structures. And I am confused why the authors cited Extended Data Fig. 8 in line 458, is there any information for structural comparison here?

Major issue:

It is a pity that, the authors haven't shown any structure comparison with other reported mycobacterial GT-C enzymes, including the ethambutol target EmbB and EmbC.

We thank the Reviewer for having raised this important point. In response, we have added a comprehensive structural comparison section that analyzes AftB in relation to other mycobacterial arabinosyltransferases, including AftA, AftD, and EmbB as a representative of the Emb protein family. We chose EmbB as a representative because it shares high structural similarity with EmbA and EmbC. This comparison is presented in a new supplementary figure (Supplementary Figs. 3 and 4).

Should this be added in the Extended Data Figures ?

In this paper the activity of AftB was detected in the presence of 10 mM MgCl₂, however they had no description on whether the activity is cation-dependent, nor had they shown an EDTA control of their AftB activity analysis. However, this is important for mechanism analysis.

The conditions used in the AraT assay were those used previously by other investigators and include MgCl₂; our focus was not to probe the metal ion dependence of the enzyme. A literature search to determine if the metal ion dependence of the Aft and Emb enzymes had been previously investigated revealed that, surprisingly, it appears that is has not been. We agree that this is useful information, but we see it as outside the scope

of the present manuscript. Importantly, while a metal ion may be involved, we do not think it would meaningfully change our key conclusion that the enzyme proceeds through a double displacement mechanism.

This question was raised based on Fig.3b and 3d, wherein the arabinosyl-phosphate group is bound in a negatively charged cavity. So how the phosphate leaving group is activated in the transfer reaction? Metal-ion dependency is a part of the mechanism analysis.

It seems that the Fab used for EM study is close to the proposed Araf4 cavity and may block the substrate entrance. However, the effect of Fab was not considered during their AraT activity study. The authors should state if the Fab-B3 inhibit the AftB enzymatic activity, and clarify whether the EM structure they determined is in the inhibitory state or not. As the Fab-bound structure was used for the following MD simulation study and the authors failed to obtain a stable Araf4-/ Araf6- bound MD result.

While our atomistic simulations with terminal-Araf_{4/6} are unstable, the CG simulations with the apo state without the Fab show clear binding. When we performed CG simulations with Araf₄ and the Fab present, there was still binding of the sugar to the proposed binding site, consistent with the fact that the Fab does not inhibit acceptor binding (Extended Data Fig.10a (ii)).

Additionally, RMSF analysis revealed that protein dynamics remain similar across apo, Fab-bound, substrate-bound, and product-bound states (Extended Data Fig.10a (i)), indicating that Fab does not significantly alter protein flexibility. We have added these analyses to the Results section (lines 236-240).

Fab does not inhibit acceptor binding is supported by in silico analysis, rather than “the fact”. From the EM data, Fab doesn’t significantly interfere with the proposed acceptor entrance.

An easy way is to measure the AftB enzymatic activity in the presence of Fab, it will be perfect to introduce Fab in the activity assay established by the authors.

Line277, we are not very convinced by the double-displacement mechanism of AftB proposed by the authors. As this is only supported by the position of the catalytic residue which is close to the donor’s anomeric carbon. However, this is weak as they lack the important acceptor-bound structure. Can the authors just rule out any other possibility of the mechanism? If not, please describe in the discussion.

We too were disappointed that we could not obtain any complexes with an acceptor bound and that the fluorinated FPA did not produce a stable enzyme-linked intermediate. Because this is the first example of a retaining GT-C enzyme to be structurally-characterized, drawing parallels with other enzymes is impossible. To address this point, we enhanced the discussion (lines 384–412) better comparing similarities to the structure of WbbB (a GT-B enzyme), which is one of two known retaining glycosyltransferases demonstrated to proceed via a double displacement mechanism and also retaining glycosidases, which proceed by a similar mechanism. It

should be appreciated that the key element of data leading to the suggestion (and now acceptance) of the alternate S_Ni mechanism for most retaining (GT-A and GT-B) glycosyltransferases, is the lack of a catalytic residue appropriately placed to act as a nucleophile. Such a residue is present in AftB, and the distance to the anomeric center in the donor is identical to a donor-bound complex of WbbB and in the same range of that typically found in retaining glycosidases.

Still not fully convinced. *Suggestion:* Combine and move the “catalytic mechanism of AftB” to the “mechanistic insights into glycosyl transfer” in the discussion.

Other:

Extended Fig. 6 e, the panel was AftA, while the caption was STT3B.

Supplementary Figure 4 c, do you mean *Mtb* AftA from PDB databank or *Ms* AftA from a predicted model? There is no reported structure of *Ms* AftA that I can find.